# MM-Embed: Universal Multimodal Retrieval with Multimodal LLMs

**Sheng-Chieh Lin** [*1,2]  **Chankyu Lee** [1]  **Mohammad Shoeybi** [1]  **Jimmy Lin** [2]

**Bryan Catanzaro** [1]  **Wei Ping** [*1]

[1] NVIDIA  [2] University of Waterloo

## Abstract

State-of-the-art retrieval models typically address a straightforward search scenario, in which retrieval tasks are fixed (e.g., finding a passage to answer a specific question) and only a single modality is supported for both queries and retrieved results. This paper introduces techniques for advancing information retrieval with multimodal large language models (MLLMs), enabling a broader search scenario, termed universal multimodal retrieval, where multiple modalities and diverse retrieval tasks are accommodated. To this end, we first study fine-tuning an MLLM as a bi-encoder retriever on 10 datasets with 16 retrieval tasks. Our empirical results show that the fine-tuned MLLM retriever is capable of understanding challenging queries, composed of both text and image, but it underperforms compared to a smaller CLIP retriever in cross-modal retrieval tasks due to the *modality bias* exhibited by MLLMs. To address the issue, we propose modality-aware hard negative mining to mitigate the *modality bias* exhibited by MLLM retrievers. Second, we propose continuously fine-tuning the universal multimodal retriever to enhance its text retrieval capability while preserving multimodal retrieval capability. As a result, our model, MM-Embed, achieves state-of-the-art performance on the multimodal retrieval benchmark M-BEIR, which spans multiple domains and tasks, while also surpassing the state-of-the-art text retrieval model, NV-Embed-v1, on the MTEB retrieval benchmark. Finally, we explore prompting the off-the-shelf MLLMs as zero-shot rerankers to refine the ranking of the candidates from the multimodal retriever. We find that, through prompt-and-reranking, MLLMs can further improve multimodal retrieval when the user queries (e.g., text-image composed queries) are more complex and challenging to understand. These findings also pave the way for advancing universal multimodal retrieval in the future. We release the model weights at: https://huggingface.co/nvidia/MM-Embed.

## 1 Introduction

Information retrieval is crucial for a variety of downstream tasks, such as question answering (Kwiatkowski et al., 2019), fact-checking (Thorne et al., 2018), and retrieval-augmented generation (Lewis et al., 2020). State-of-the-art retrievers often focus on narrow scenarios. For example, LLM-based retrievers (Wang et al., 2023; Lee et al., 2024; Meng et al., 2024) are limited to text-to-text retrieval tasks, in which both the query and the retrieved results are text-only. Recent work on multimodal retrieval (Zhang et al., 2024; Jiang et al., 2024) focuses on specific tasks and assumes a homogeneous document format. However, in real-world applications, documents and queries often incorporate diverse formats or modalities, such as text, images, and interleaved text-image content. To advance information retrieval and support broader search scenarios, this work explores leveraging multimodal LLMs (MLLMs; Liu et al., 2023a; 2024; Dai et al., 2024) for universal multimodal

---

*Sheng-Chieh Lin did this work during an internship at NVIDIA. Correspondence to: Sheng-Chieh Lin ⟨s269lin@uwaterloo.ca⟩, Wei Ping ⟨wping@nvidia.com⟩.

retrieval, accommodating diverse user-instructed tasks with multimodal queries and documents, as illustrated in Figure 1.

We first explore fine-tuning MLLM-based bi-encoder retrievers with instructions as a guide (Asai et al., 2023) on 16 multimodal retrieval tasks from M-BIER (Wei et al., 2023). We find that MLLM-based retrievers significantly outperform CLIP-based retrievers in the challenging tasks that involve interleaved text–image queries, such as visual question answering and composed image retrieval (tasks 3 and 7 in Figure 1). However, MLLM-based retrievers underperform in cross-modal retrieval tasks due to the *modality bias* exhibited by MLLMs. That is, given a text-based query with the instruction to retrieve an image (e.g., task 9 in Figure 1), an MLLM-based retriever tends to retrieve a relevant text-only document rather than one containing images, especially when we enhance the retriever's text retrieval capability. To address the issue, we propose modality-aware hard negative mining (Section 4.1.1) and continuously fine-tuning for text-to-text retrieval (Section 4.1.2). Our final retriever, coined MM-Embed, is the first universal multimodal retriever to achieve state-of-the-art performance while maintaining competitive text-to-text retrieval across diverse tasks.

Finally, we explore prompting MLLMs as zero-shot rerankers. Surprisingly, we find that the zero-shot MLLM-based rerankers can further boost retrieval accuracy in the tasks where user queries are interleaved text-image and particularly challenging to understand. For example, in the composed image retrieval dataset, CIRCO (Baldrati et al., 2023), the zero-shot reranker can refine the ranked lists and significantly boost the accuracy (mAP@5) over 7 points from the existing state-of-the-art composed-image retriever (Zhang et al., 2024) and our universal multimodal retrievers. This finding indicates that there is still room for improvement in such challenging tasks to advance universal multimodal retrieval. Moreover, knowledge distillation from zero-shot or few-shot MLLM-based rerankers to retrievers is a promising direction.

We summarize our contributions as follows: *i)* We present a study on applying MLLMs to universal multimodal retrieval. *ii)* We are the first to develop MLLM-based universal multimodal retrievers. Notably, our MM-Embed, initialized from the existing best-performing text retriever (NV-Embed-v1; Lee et al., 2024), not only achieves state-of-the-art results in the universal multimodal retrieval benchmark M-BEIR (Wei et al., 2023), but also surpasses NV-Embed-v1 in text-to-text retrieval tasks on MTEB. *iii)* We explore prompting MLLMs as zero-shot rerankers in various multimodal retrieval tasks. Surprisingly, we find that zero-shot MLLM-based rerankers can improve ranking accuracy over strong retrievers in challenging tasks involving interleaved text-image queries.

We organize the rest of the paper as follows: We discuss related work in § 2. We introduce the definition of universal multimodal retrieval in § 3 and present the proposed method in § 4. We report experimental results in § 5 and conclude the paper in § 6.

## 2    RELATED WORK

**Instruction-Aware Dense Representation Learning.**    Asai et al. (2023) is the first study to identify the implicit search intent behind each retrieval task and proposes fine-tuning a retriever to learn diverse retrieval tasks with handwritten task instructions. Su et al. (2023) and existing state-of-the-art LLM-based text embedding models (Wang et al., 2023; Meng et al., 2024; Lee et al., 2024) adopt this approach to broader tasks beyond text retrieval, such as text classification and clustering. Recently, Wei et al. (2023) propose a universal multimodal retrieval dataset, M-BEIR, and find that instruction-aware dense retrieval fine-tuning is crucial to tackle universal multimodal retrieval.

**Vision-Language Models for Multimodal Retrieval.**    With the advancement of pre-trained vision-language models (Radford et al., 2021; Li et al., 2022), the research focus shifts from single-modal (Bajaj et al., 2016; Fu et al., 2023) to cross-modal (Lin et al., 2014; Han et al., 2017; Liu et al., 2021a) or more complex multimodal retrieval tasks (Liu et al., 2021b; Wu et al., 2021; Baldrati et al., 2023). However, the aforementioned tasks assume a homogeneous modality for queries and documents, thus limiting their application. Liu et al. (2023c) take one step further to tackle the retrieval scenario involving a candidate pool with heterogeneous modalities but still limit itself to a single retrieval task.

Wei et al. (2023) extend the study to a more general scenario, where retrievers are required to deal with queries, a candidate pool with heterogeneous modalities and diverse retrieval tasks. However,

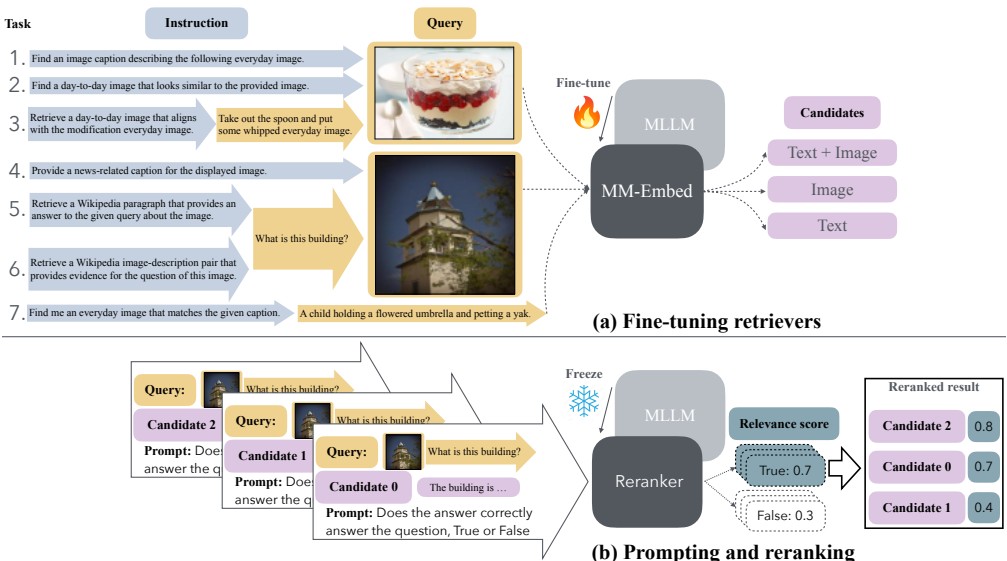

Figure 1: Illustration of universal multimodal retrieval in (a), where each task consists of a task-specific instruction and query. Both queries and candidate documents are in heterogeneous formats (i.e., text, image or, interleaved text-image). In this work, we explore (a) fine-tuning MLLM-based universal multimodal retrievers and (b) prompting pre-trained MLLMs for zero-shot reranking over retrieved candidates. We adopt LLaVa-Next (Liu et al., 2024) as our MLLM backbone.

the study is limited to CLIP-based retrievers and ignores important text-to-text retrieval tasks, such as fact-checking (Thorne et al., 2018) and entity retrieval (Hasibi et al., 2017). While Koukounas et al. (2024) aim to fine-tune a CLIP-based retriever with both strong text-to-text and multimodal retrieval capabilities, they only consider simple multimodal retrieval tasks: image-caption retrieval (Young et al., 2014; Lin et al., 2014). Concurrent with our work, Jiang et al. (2024) propose to fine-tune MLLMs on the NLI dataset (Bowman et al., 2015) and demonstrate their transferability to multimodal retrieval. In this paper, we are the first to study how to fine-tune an MLLM-based universal multimodal retriever while maintaining strong text-to-text retrieval capability.

**Prompting Multimodal LLMs for Reranking.** Instruction tuning has enabled large language models (LLMs) to tackle a wide range of tasks in a zero-shot setting. Building on this, prior studies have investigated prompting LLMs for text reranking (Ma et al., 2023; Sun et al., 2023; Zhuang et al., 2024b). In this work, we extend this line of research into multimodal LLMs, exploring their potential as zero-shot rerankers for multimodal tasks. Notably, Qu et al. (2024) introduce a framework using multimodal LLMs for zero-shot reranking through a generative retrieval approach (Li et al., 2024). However, their method is constrained to retrieval tasks with text-only queries. In contrast, our approach broadens the scope by prompting multimodal LLMs to handle diverse multimodal reranking tasks, accommodating queries and documents that can be in text, image, or interleaved text–image formats. This generalization enables versatile applications in multimodal ranking settings.

## 3 UNIVERSAL MULTIMODAL RETRIEVAL

Following the framework of Lin et al. (2021), we formulate the task of retrieval as follows: given a query $q$, the goal is to retrieve a ranked list of candidates $\{c_1, c_2, \cdots c_k\} \in C$ to maximize some ranking metrics such as nDCG, where $C$ is the collection of documents. In this work, we borrow the setting of universal multimodal retrieval from Wei et al. (2023), where user queries and candidates may consist of text, image or interleaved text-image; i.e., $q \in \{q^{\text{txt}}, q^{\text{img}}, (q^{\text{txt}}, q^{\text{img}})\}$; $c \in \{c^{\text{txt}}, c^{\text{img}}, (c^{\text{txt}}, c^{\text{img}})\}$. Additionally, there are multiple search intents behind a search query, which can be elaborated through task-specific instructions (Asai et al., 2023). For example, in tasks 1 and 2 of Figure 1, given the same image as a query, the search intent is to find an image caption and similar image, respectively. Thus, in universal multimodal retrieval, given a multimodal query and task instruction $inst$, we aim to retrieve a list of candidates from a pool of multimodal documents to

maximize a specified ranking metric. Note that we only consider text and image in this work while other modalities, such as audio and video can be included, which we leave for future work.

## 4 METHOD

In this section, we describe our approach to universal multimodal retrieval by leveraging multimodal LLMs (MLLMs), specifically LLaVa-Next (Liu et al., 2024). In Section 4.1, we first fine-tune an MLLM-based retriever to project multimodal user queries, along with task instructions, into the same semantic space as multimodal documents, enabling $k$-nearest neighbor search (Johnson et al., 2021). In Section 4.2, we present our method to use MLLMs to rerank the top-$k$ candidates retrieved by the universal multimodal retriever.

### 4.1 FINE-TUNING MULTIMODAL LLMS FOR UNIVERSAL MULTIMODAL RETRIEVAL

We fine-tune an MLLM-based retriever parameterized by $\theta$ (i.e., $\eta^\theta$) under the guidance of task-specific instructions, aiming to capture the implicit intent behind retrieval tasks. Specifically, given a user query $q_i$ with the specified task instruction $inst_i$ and its corresponding relevant candidate, $c_i^+$, we minimize the contrastive loss (Gutmann & Hyvärinen, 2010):

$$L = -\frac{1}{|\mathcal{B}|} \sum_{i=1}^{|\mathcal{B}|} \log \frac{\exp\left(\eta^\theta(inst_i, q_i) \cdot \eta^\theta(c_i^+)/\tau\right)}{\sum_{c' \in \mathcal{D}} \exp(\eta^\theta(inst_i, q_i) \cdot \eta^\theta(c')/\tau)}, \qquad (1)$$

where $\eta^\theta(\cdot) \in \mathbb{R}^d$ is a normalized vector and $\tau$ is the temperature scaling factor. Ideally, $\mathcal{D}$ includes all the candidate documents. However, including all the candidate documents is not computationally feasible; thus, mining informative negative candidates as an alternative to $\mathcal{D}$ is crucial for successful contrastive learning. In this work, we propose modality-aware negative mining for contrastive learning in the scenario for universal multimodal retrieval.

#### 4.1.1 MODALITY-AWARE HARD NEGATIVE MINING

Prior work (Karpukhin et al., 2020; Xiong et al., 2021; de Souza P. Moreira et al., 2024) has demonstrated that hard negative mining significantly improves representation learning for text-to-text retrieval. In the traditional retrieval setting, where the corpus consists of documents with a homogeneous modality, a document is considered a hard negative if it lacks the required information but is still retrieved by a model. However, in the scenario of universal multimodal retrieval, where the corpus contains documents involving diverse modalities, a user's desired modality as specified in task instructions (i.e., text, image or interleaved text-image) should be taken into consideration. For example, as shown in Figure 1, the first and second users issue the same query along with different instructions, requiring the documents in text and image formats, respectively. To address this, we propose modality-aware hard negative mining to guide models in retrieving candidates that meet both the users' information needs and their preferred modality.

Specifically, we first fine-tune an MLLM-based retriever by using in-batch samples as random negatives; i.e., $\mathcal{D} = (c_1^+, \cdots, c_{|\mathcal{B}|}^+)$ in Eq. (1). The candidate documents in the mini-batch, except $c_i^+$, are considered random negatives for $(inst_i, q_i)$. The fine-tuned model is denoted as $M^{\text{rand}}$. For each query $q_i$ and its associated instruction $inst_i$ in the training set, we generate two types of negative samples from the top-50 candidates retrieved by $M^{\text{rand}}$: *i)* negatives with an incorrect modality ($C_i^1$), where the candidate ranks higher than the labeled positive but has a different modality from the desired one, and *ii)* negatives with unsatisfactory information ($C_i^2$), where the candidate ranks lower than position $k'$ but has the same desired modality.

Previous studies on text retrieval (Xiong et al., 2021; de Souza P. Moreira et al., 2024) have shown that setting $k'$ to a small number may introduce false positives while setting $k'$ to a large number could make the negative samples too easy. In our experiment, we set $k' = 45$ following the state-of-the-art text retrieval training in Lin et al. (2023). During training, given the query $q_i$ with the associated instruction $inst_i$, we generate a triplet, $((inst_i, q_i), c_i^+, c_i^-)$, by sampling a hard negative $c_i^-$ from either $C_i^1$ or $C_i^2$ with the same probability; i.e., $\mathcal{D} = (c_1^+, c_1^-, \cdots, c_{|\mathcal{B}|}^+, c_{|\mathcal{B}|}^-)$ in Eq. (1). Thus, in the setting of hard negative mining, the negatives mined for $(inst_i, q_i)$ include *i)* the hard

negative $c_i^-$ and *ii)* all positives and hard negatives from other queries, which are considered random negatives. Note that the setting of hard negative mining includes twice as many candidate documents as random negative mining under the same batch size $|\mathcal{B}|$. For a fair comparison, we use $2 \cdot |\mathcal{B}|$ and $|\mathcal{B}|$ when fine-tuning with random and hard negatives, respectively. Figure 2 in the Appendix showcases both types of negative samples. We observe that the negatives from $C_1$ are sentences that are semantically similar to the queries but do not match the user's desired modality.

### 4.1.2 CONTINUOUS TEXT-TO-TEXT RETRIEVAL FINE-TUNING

Since text-to-text retrieval remains one of the most widely used retrieval tasks, we further fine-tune $M^{\text{hard}}$ on a diverse set of public text-to-text retrieval tasks, including MS MARCO (Bajaj et al., 2016), HotpotQA (Yang et al., 2018), Natural Question (Kwiatkowski et al., 2019), PAQ (Lewis et al., 2021), StackExchange (Stack-Exchange-Community., 2023), Natural Language Inference (Bowman et al., 2015), SQuAD (Rajpurkar et al., 2016), ArguAna (Wachsmuth et al., 2018), BioASQ (Nentidis et al., 2023), FiQA (Maia et al., 2018), and FEVER (Thorne et al., 2018). As these datasets do not contain negative samples, we employ the fine-tuned LLM-based retriever (NV-Embed-v1; Lee et al., 2024) to mine hard negatives in our experiments (see de Souza P. Moreira et al. (2024) for details).

During continuous fine-tuning, we uniformly sample triplets from both universal multimodal and text-to-text retrieval training data. Note that for each query $q_i$ in the universal multimodal retrieval training data, we use $M^{\text{hard}}$ to mine second-type hard negatives $C_i^2$ again. Since no first-type hard negatives (i.e., $C_i^1 = \emptyset$) are found by $M^{\text{hard}}$, we retain the first-type hard negatives mined by $M^{\text{rand}}$.

### 4.2 PROMPTING MULTIMODAL LLMS FOR RERANKING

Prior work (Sun et al., 2023; Jin et al., 2024) has demonstrated that instruction-fine-tuned LLMs can be prompted to rerank candidates in text-to-text retrieval tasks. In this work, we directly prompt pre-trained LLaVa-Next (i.e., the same MLLM backbone for retrievers but without fine-tuning) to further rerank the top 10 retrieved candidates from universal multimodal retrievers. Following the approach of Nogueira et al. (2020), we frame the reranking task as a series of True/False questions. Specifically, given a query and a retrieved candidate, we prompt LLaVa-Next to determine whether the retrieved candidate satisfies the query by answering "True" or "False". For example, in the image caption retrieval (task 1 in Figure 1), given an image query, $q^{img}$, and a retrieved text-based candidate, $c^{txt}$, we use the following prompt: "$< q^{img} >\backslash nCaption:< c^{txt} >\backslash nDoes\ the\ above\ daily\text{-}life\ image$ *match the caption? True or False*". Additionally, in visual question answering retrieval (task 5 in Figure 1), given a visual question, *<Qry image><Qry text>*, and a retrieved text-based candidate, *<Doc text>*, we use the following prompt: *<Qry image>\nQuestion:<Qry text>\nAnswer:<Doc text>\nDoes the answer correctly answer the question? True or False*. We refer readers to Table 18 in Appendix for the specific prompts used in different multimodal retrieval tasks.

To compute relevance scores, we apply the `Softmax` operation to the logits of the "True" and "False" tokens, and use the probability of the "True" token as the relevance score for reranking. Our preliminary study in Section 5.3.3 shows that zero-shot MLLM-based rerankers mainly improve the tasks where queries are interleaved text and images, such as composed image retrieval and visual question answering, as shown in the tasks 3, 5, and 6 of Figure 1.

## 5 EXPERIMENTS

### 5.1 DATASETS AND MODELS

**Multimodal Retrieval Dataset.** We evaluate models' universal multimodal retrieval capabilities using the M-BEIR dataset (Wei et al., 2023), which is constructed from 10 datasets with 16 diverse multimodal retrieval tasks across 4 domains, as listed in Appendix Table 10.[1] We train our models on the M-BEIR 1.1M training queries and evaluate their effectiveness on the 190K test queries. Following the global evaluation setting of the M-BEIR dataset, for each query, candidates are retrieved from a unified candidate pool of 5.6M multimodal documents spanning all 10 datasets. We report the average Recall@5 (R@5) as the retrieval accuracy across all test queries in each dataset, except

---

[1]`https://huggingface.co/datasets/TIGER-Lab/M-BEIR`

for Fashion200K and FashionIQ, for which we report Recall@10 (R@10). We refer readers to Wei et al. (2023) for more details on the construction of the M-BEIR dataset.

**Text-to-Text Retrieval Dataset.** While M-BEIR contains the WebQA dataset for text-to-text retrieval evaluation, we conduct a more comprehensive text-to-text retrieval evaluation using the MTEB dataset (Muennighoff et al., 2023). Specifically, we evaluate our models on 15 diverse text retrieval datasets.[2] Following the established procedure, we report the average nDCG@10 across the 15 text retrieval datasets. Note that, unlike in M-BEIR, where candidates are retrieved from a unified pool across all tasks, in the MTEB retrieval tasks, candidates are retrieved from separate corpora for each task.

**Backbone Model Choices.** In this work, we utilize two representative backbones of vision-language models to build universal multimodal retrievers, CLIP (Radford et al., 2021) and LLaVa-Next (Liu et al., 2024). For CLIP, we initialize from the CLIP-Large model and employ the best-performing modeling approach from Wei et al. (2023), denoted as $\text{CLIP}_{\text{SF}}$.[3] This method fuses the input image and text features by separately encoding each input (query or document) image and text into distinct vectors, which are then summed to create a fused vector (Liu et al., 2023c). Additionally, we report results for BLIP (Li et al., 2022), which integrates text information into the image encoder through cross-attention. We use $\text{BLIP}_{\text{FF}}$ from Wei et al. (2023), which is fine-tuned on the M-BEIR dataset with random negative.[4]

LLaVa-Next (Liu et al., 2024) is a multimodal LLM (MLLM), which integrates a CLIP image encoder, LLM and a vision-language MLP projector to align image features to the input embedding space of the LLM. We use LLaVa-Next with the Mistral 7B (Jiang et al., 2023) as the backbone LLM.[5] We experiment with three variants: (1) LLaVa-E: the $<eos>$ token embedding is used to aggregate information from the multimodal input, a method commonly employed in prior work on text retrieval (Wang et al., 2023; Ma et al., 2024b); (2) LLaVa-P: the MLLM is prompted to summarize each multimodal query (or document) input in one word, using embedding of the last token to encode multimodal input;[6] (3) NVEmb: The LLM from LLaVa-Next is replaced by the fine-tuned LLM-based text retrieval model NV-Embed-v1 (Lee et al., 2024) while all other components (i.e., image encoder and vision-language MLP projector) remain unchanged.[7] Note that the backbone of NV-Embed-v1 is also Mistral 7B. The instructions for LLaVa-E (or NVEmb) and LLaVa-P are illustrated in Appendix Table 16 and 17, respectively. For reranking experiments, we also utilize LLaVa-Next with Mistral 7B, and the prompts are listed in Appendix Table 18.

**Retriever Training Details.** For each backbone, we start by fine-tuning $M^{\text{rand}}$ with random negatives; i.e., $\mathcal{D} = (c_1^+, \cdots, c_{|\mathcal{B}|}^+)$ in Eq. (1). The fine-tuned model is denoted $M^{\text{rand}}$. For the CLIP backbone, following (Wei et al., 2023), we fine-tune $\text{CLIP}_{\text{SF}}$ for 20 epochs with a learning rate of $1e-5$. For LLaVa-Next backbone, we fine-tune the models for 2 epochs with a learning rate of $1e-4$. Note that for LLaVa-Next backbone, we only fine-tune the vision-language projector and LoRA ($r = 8, \alpha = 64$) added to the language model. At the stage of fine-tuning $M^{\text{hard}}$ with hard negatives, we mine the two types of hard negatives following Section 4.1.1 using each retriever. Then, we fine-tune each retriever using its own mined hard negatives with the same training procedure as the first stage; i.e., $\mathcal{D} = (c_1^+, c_1^-, \cdots, c_{|\mathcal{B}|}^+, c_{|\mathcal{B}|}^-)$ in Eq. (1). We fine-tune the models with the batch size of $128 \times 8$ and $64 \times 8$ when using random and hard negatives, respectively. When GPU memory is insufficient for the designated batch size, we use gradient accumulation. Note that when fine-tuning $M^{\text{hard}}$, we initialize the models using the pre-trained model, rather than continuously fine-tuning $M^{\text{rand}}$. We denote the models fine-tuned with random and hard negatives as $M^{\text{rand}}(\cdot)$ and $M^{\text{hard}}(\cdot)$, respectively. We refer readers to Appendix A.1 for more details.

---

[2]The 15 retrieval datasets in MTEB are derived from public datasets in BEIR (Thakur et al., 2021), excluding BioASQ, Signal-1M, TREC-NEWS, Robust04.

[3]`https://huggingface.co/openai/clip-vit-large-patch14`

[4]`https://huggingface.co/TIGER-Lab/UniIR/blob/main/checkpoint/BLIP_FF/blip_ff_large.pth`

[5]`https://huggingface.co/llava-hf/llava-v1.6-mistral-7b-hf`

[6]We refer readers to Appendix Table 17 for the prompt and more details from the prior work (Zhuang et al., 2024a; Jiang et al., 2024).

[7]`https://huggingface.co/nvidia/NV-Embed-v1`

Table 1: Main results on retrieval. Following Wei et al. (2023), we report R@5 for all the datasets, except for Fashion200K and FashionIQ, for which we report R@10. The tasks of single-modal and multi-modal queries are tasks 1–5 and 6–8, respectively. For MTEB text retrieval, we report nDCG@10 averaged across 15 retrieval tasks (detailed in Appendix Table 15). See more comparison in Appendix Table 12 and 13.

| Task | Dataset | $M^{\text{rand}}$ | | | | | $M^{\text{hard}}$ | | | MM-Embed |
|---|---|---|---|---|---|---|---|---|---|---|
| | | CLIP$_{\text{SF}}$ | BLIP$_{\text{FF}}$ | LLaVa-E | LLaVa-P | NVEmb | CLIP$_{\text{SF}}$ | LLaVa-P | NVEmb | |
| 1. $q^{\text{txt}} \rightarrow c^{\text{img}}$ | VisualNews | 43.8 | 23.0 | 33.2 | 34.2 | 32.1 | 42.7 | 39.7 | 41.1 | 41.0 |
| | MSCOCO | 72.0 | 75.6 | 69.3 | 70.8 | 64.6 | 69.2 | 73.8 | 72.7 | 71.3 |
| | Fashion200K | 16.4 | 25.4 | 13.5 | 13.3 | 10.4 | 19.7 | 17.4 | 18.6 | 17.1 |
| 2. $q^{\text{txt}} \rightarrow c^{\text{txt}}$ | WebQA | 83.2 | 79.5 | 88.6 | 88.8 | 92.1 | 88.2 | 93.6 | 95.6 | 95.9 |
| 3. $q^{\text{txt}} \rightarrow (c^{\text{img}}, c^{\text{txt}})$ | EDIS | 46.5 | 50.3 | 55.9 | 56.6 | 55.1 | 54.2 | 68.8 | 69.8 | 68.8 |
| | WebQA | 76.0 | 79.7 | 80.3 | 81.6 | 81.3 | 80.1 | 84.9 | 84.8 | 85.0 |
| 4. $q^{\text{img}} \rightarrow c^{\text{txt}}$ | VisualNews | 39.5 | 21.1 | 32.4 | 33.3 | 30.4 | 40.6 | 39.4 | 41.4 | 41.3 |
| | MSCOCO | 91.0 | 88.8 | 91.8 | 92.2 | 90.3 | 88.5 | 89.5 | 88.9 | 90.1 |
| | Fashion200K | 17.2 | 27.6 | 13.9 | 14.7 | 13.2 | 20.0 | 17.5 | 19.9 | 18.4 |
| 5. $q^{\text{img}} \rightarrow c^{\text{img}}$ | NIGHTS | 31.6 | 33.0 | 31.8 | 30.7 | 30.4 | 31.9 | 31.8 | 31.1 | 32.4 |
| 6. $(q^{\text{img}}, q^{\text{txt}}) \rightarrow c^{\text{txt}}$ | OVEN | 40.4 | 38.7 | 37.9 | 39.1 | 36.3 | 40.9 | 42.9 | 42.6 | 42.1 |
| | InfoSeek | 26.1 | 19.7 | 31.0 | 32.9 | 33.3 | 27.6 | 37.2 | 35.8 | 42.3 |
| 7. $(q^{\text{img}}, q^{\text{txt}}) \rightarrow c^{\text{img}}$ | FashionIQ | 24.2 | 28.5 | 27.4 | 27.0 | 26.0 | 21.7 | 25.8 | 26.6 | 25.7 |
| | CIRR | 43.2 | 51.4 | 48.1 | 45.4 | 45.3 | 38.3 | 49.5 | 50.8 | 50.0 |
| 8. $(q^{\text{img}}, q^{\text{txt}}) \rightarrow (c^{\text{img}}, c^{\text{txt}})$ | OVEN | 60.9 | 57.8 | 61.6 | 62.6 | 61.7 | 61.6 | 63.9 | 63.5 | 64.1 |
| | InfoSeek | 45.9 | 27.7 | 50.3 | 50.0 | 53.4 | 47.1 | 54.4 | 53.5 | 57.7 |
| M-BEIR Avg. | All | 47.4 | 45.5 | 47.9 | 48.3 | 47.2 | 48.3 | 51.9 | 52.3 | **52.7** |
| | Single-modal Qry | 51.7 | 50.4 | 51.0 | 51.6 | 50.0 | 53.5 | 55.6 | **56.4** | 56.1 |
| | Multi-modal Qry | 40.1 | 37.3 | 42.7 | 42.8 | 42.7 | 39.5 | 45.6 | 45.5 | **47.0** |
| MTEB (Muennighoff et al., 2023) Text Retrieval Avg. | | - | - | - | 40.8 | 51.6 | - | 46.4 | 49.7 | **60.3**[*] |

[*] ranked top-5 on MTEB retrieval task leaderboard. NV-Embed-v1 (Lee et al., 2024) scores 59.36 in MTEB retrieval task.

To enhance text-to-text retrieval capability, we continuously fine-tune $M^{\text{hard}}$(NVEmb) with a learning rate of $2e-5$ using a mixture of training data from M-BEIR and public text retrieval datasets mentioned in Section 4.1.2 for 4.5K steps. The final model is called MM-Embed.

## 5.2 Main Results

**Universal Multimodal Retrieval.** Table 1 presents the retrieval accuracy of different retrievers. In the M-BEIR evaluation, we observe that when fine-tuning with random negatives, LLaVa-P achieves the highest overall retrieval effectiveness. This result indicates that LLaVa-P effectively aggregates multimodal input information into a single word representation. While MLLM-based retrievers outperform CLIP$_{\text{SF}}$ on tasks involving multimodal queries, they still lag behind CLIP$_{\text{SF}}$ on tasks with single-modal queries, especially in cross-modality retrieval (i.e., tasks 1 and 4). In addition, NVEmb achieves the best text-to-text retrieval accuracy on WebQA Task 2. It is worth noting that although BLIP$_{\text{FF}}$ performs the worst overall, it demonstrates notably strong performance in the fashion domain (e.g., Fashion200K and FashionIQ) but performs worse in news domain (e.g., VisualNews), likely due to differences in the text–image pairs used for pre-training between CLIP and BLIP.

Observing the models fine-tuned with hard negatives, MLLM-based retrievers show significant improvements in retrieval accuracy, particularly in tasks involving single-modal queries. On the other hand, CLIP$_{\text{SF}}$ does not show a similar improvement. This could be attributed to the fact that CLIP has been well pre-trained for cross-modal retrieval, whereas MLLM-based retrievers, fine-tuned with a contrastive learning objective for only 2 epochs, may still be underfitting. Fine-tuning with hard negatives accelerates the contrastive learning process of MLLM-based retrievers.

Table 2 reveals another factor contributing to the lower retrieval accuracy of MLLM-based retrievers for single-modal queries: text retrieval bias. This issue is particularly obvious for NVEmb. We compare models' retrieval accuracy on text-image and image-text retrieval (tasks 1 and 4) on MSCOCO. The compar-

Table 2: Retrieval analysis on MSCOCO. M.A.@1 denotes the modality accuracy of the top-1 candidate. More results of M.A.@1 are presented in Appendix Table 14.

| Task | Metric | $M^{\text{rand}}$ | | | | $M^{\text{hard}}$ | | |
|---|---|---|---|---|---|---|---|---|
| | | CLIP$_{\text{SF}}$ | LLaVa-E | LLaVa-P | NVEmb | CLIP$_{\text{SF}}$ | LLaVa-P | NVEmb |
| 1. | R@1 | 42.6 | 33.9 | 41.7 | 14.1 | 45.8 | 50.7 | 49.8 |
| | R@5 | 72.0 | 69.3 | 70.8 | 64.6 | 69.2 | 73.8 | 72.7 |
| | M.A.@1 | 92.6 | 79.9 | 91.0 | 42.1 | 98.3 | 100.0 | 100.0 |
| 4. | R@1 | 72.3 | 73.0 | 73.4 | 69.3 | 63.8 | 72.7 | 72.4 |
| | R@5 | 91.0 | 91.8 | 92.2 | 90.3 | 88.5 | 89.5 | 88.9 |
| | M.A.@1 | 98.7 | 99.2 | 99.8 | 96.3 | 94.2 | 100.0 | 100.0 |

Table 3: Results of zero-shot reranking on tasks 6–8 from the M-BEIR dataset.

| Task | Dataset | $M^{\text{hard}}$ (NVEmb) | | MM-Embed | |
|---|---|---|---|---|---|
| | | Retrieval | Rerank | Retrieval | Rerank |
| 6. | OVEN | 42.6 | 44.3 | 42.1 | 43.5 |
| | InfoSeek | 35.8 | 37.1 | 42.3 | 43.1 |
| 7. | FashionIQ | 26.6 | 20.0 | 25.7 | 19.0 |
| | CIRR | 50.8 | 48.6 | 50.0 | 48.2 |
| 8. | OVEN | 63.5 | 65.8 | 64.1 | 65.9 |
| | InfoSeek | 53.5 | 54.5 | 57.7 | 57.3 |

Table 4: Results of zero-shot reranking on composed image retrieval task, CIRCO (Baldrati et al., 2023).

| Retrieval Model | Retrieval | Rerank |
|---|---|---|
| MagicLens (Zhang et al., 2024) | 24.9 | 32.4 |
| E5-V (Jiang et al., 2024) | 19.1 | 31.0 |
| $M^{\text{rand}}$(CLIP$_{\text{SF}}$) | 12.7 | 31.6 |
| BLIP$_{\text{FF}}$ (Wei et al., 2023) | 26.6 | 36.1 |
| $M^{\text{hard}}$ (LLaVa-P) | 29.0 | 37.9 |
| $M^{\text{hard}}$ (NVEmb) | 32.4 | 40.9 |
| MM-Embed | 32.3 | 39.9 |

ison shows that $M^{\text{rand}}$(LLaVa-E) and $M^{\text{rand}}$(NVEmb) exhibit significantly lower modality accuracy (M.A.@1) than $M^{\text{rand}}$(CLIP$_{\text{SF}}$) in the text-to-image retrieval task. Most incorrectly retrieved top-1 candidates from the MLLM-based retrievers are relevant texts rather than images (see Appendix Figure 2). This result indicates that MLLM-based retrievers have a bias toward relevant text rather than images, a phenomenon not caused by random sampling, as evidenced in Appendix Table 11. This issue can be mitigated by our proposed modality-aware hard negative mining.

Finally, we observe that $M^{\text{hard}}$(NVEmb) performs worse in text-to-text retrieval tasks compared to $M^{\text{rand}}$(NVEmb) but still outperforms $M^{\text{hard}}$(LLaVa-P) (i.e., WebQA task 2 and MTEB).[8] However, compared to the original NVEmb (Lee et al., 2024), its score on MTEB retrieval tasks drops by almost 10 points. After continuous fine-tuning (detailed in Section 4.1.2), the final model, MM-Embed, not only surpasses NVEmb in MTEB but also retains strong multimodal retrieval capability. We attribute the improvement in text-to-text retrieval to the effective hard negatives mined by NV-Embed-v1 as mentioned in Section 4.1.2. Notably, continuous fine-tuning significantly enhances multimodal retrieval performance in InfoSeek (col 8 vs 7 in Table 1), highlighting its effectiveness in improving the model's ability to handle knowledge-intensive multimodal retrieval tasks.

**Zero-Shot Reranking.** Table 3 reports the reranked results of the top-10 retrieved candidates from $M^{\text{hard}}$(NVEmb) and MM-Embed on the tasks involving multimodal queries. We observe accuracy improvements in visual question answering retrieval tasks (i.e., OVEN and InfoSeek), but no improvement in composed image retrieval tasks (i.e., FashionIQ and CIRR). However, as shown in Appendix Table 10, FashionIQ and CIRR, compared to OVEN and InfoSeek, only have one relevance label per query. We hypothesize that there may be additional relevant positive samples that are not labeled. We refer the reader to Appendix Figure 3 for case studies.

We conduct experiments on the CIRCO (Baldrati et al., 2023) validation set, a composed image retrieval dataset with high-quality human annotations, consisting of 219 queries and 123K candidates in total. On average, 4.2 positive samples are labeled by humans per query. Table 4 presents mAP@5 for various retrievers and their reranking results. We directly use the models and code provided by the authors to obtain the results of MagicLens (Zhang et al., 2024)[9] and E5-V (Jiang et al., 2024)[10] retrievers. For our M-BEIR fine-tuned retrievers–$M^{\text{rand}}$(CLIP$_{\text{SF}}$), $M^{\text{hard}}$(LLaVa-P), $M^{\text{hard}}$(NVEmb) and MM-Embed–we directly use the same instructions as CIRR in M-BEIR for query encoding. First, we observe that our MLLM-based retrievers outperform MagicLens and E5-V. More importantly, reranking the top-10 retrieved candidates from different retrievers significantly improves mAP@5 by at least 7 points. This result demonstrates the effectiveness of prompting an MLLM as a reranker in composed image retrieval tasks.

### 5.3 ABLATION STUDIES

#### 5.3.1 IS FINE-TUNING WITH INSTRUCTION NECESSARY?

We fine-tune NVEmb with random negatives on the M-BEIR subtasks listed in Table 5 and evaluate the models' retrieval accuracy on the development queries from each subtask. Note that, for

---

[8]We hypothesize that the decline of $M^{\text{hard}}$(NVEmb) in text-to-text retrieval tasks results from the mitigation of text retrieval bias after fine-tuning with modality-aware hard negatives.

[9]https://github.com/google-deepmind/magiclens

[10]https://github.com/kongds/E5-V

Table 5: Ablation study on fine-tuning NVEmb w/o (✗) and w/ (✓) instructions.

| Task | Dataset | zero-shot | | | | fine-tuning | |
|---|---|---|---|---|---|---|---|
| | | CLIP | LLaVa-P | NVEmb | | NVEmb | |
| | | ✗ | ✗ | ✗ | ✓ | ✗ | ✓ |
| 1. $q^{\text{txt}} \rightarrow c^{\text{img}}$ | VisualNews | 40.9 | 11.7 | 15.3 | 17.4 | 33.1 | 38.7 |
| | MSCOCO | 55.4 | 58.1 | 64.2 | 59.9 | 76.7 | 82.8 |
| | Fashion200K | 8.9 | 2.4 | 4.2 | 3.2 | 12.3 | 15.6 |
| 4. $q^{\text{img}} \rightarrow c^{\text{txt}}$ | VisualNews | 42.0 | 6.3 | 6.5 | 5.9 | 29.3 | 37.2 |
| | MSCOCO | 79.6 | 66.8 | 70.6 | 68.2 | 88.9 | 93.0 |
| | Fashion200K | 7.7 | 2.9 | 4.0 | 3.6 | 12.0 | 16.8 |
| 5. $q^{\text{img}} \rightarrow c^{\text{img}}$ | NIGHTS | 25.4 | 28.4 | 29.3 | 27.7 | 31.6 | 30.9 |

simplicity, we encode only the dataset-specific corpus containing documents of the targeted modality. For example, when evaluating retrieval accuracy for VisualNews Task 1, we encode the 542K images from VisualNews (see Appendix Table 10) into the index rather than the entire 5.6M documents from M-BEIR. We also report the CLIP and LLaVa-P (w/o instruction) zero-shot retrieval effectiveness as a reference point.[11]

From Table 5, we observe that NVEmb, as a zero-shot MLLM-based retriever, outperforms LLaVa-P and even competes with CLIP in the tasks in tasks within the Miscellaneous domain (i.e., MSCOCO and NIGHTS). This result indicates that a fine-tuned MLLM-based text retriever is capable of performing multimodal retrieval tasks (the same finding as in (Jiang et al., 2024)). Although incorporating task instructions with queries degrades the model's retrieval effectiveness (col 4 vs 3), the model fine-tuned with instructions significantly outperforms the one fine-tuned without instructions (col 6 vs 5). This indicates that task instructions can help elicit a model's task- or domain-specific knowledge for diverse multimodal retrieval tasks.

### 5.3.2 EFFECTIVENESS OF CONTINUOUS TEXT-TO-TEXT RETRIEVAL FINE-TUNING

In this section, we study the best strategy to enhance a model's capabilities in both multimodal and text-to-text retrieval. We begin by fine-tuning NVEmb on training data for both universal multimodal retrieval and text-to-text retrieval (detailed in Section 4.1.2) for 2K steps. As shown in Table 6, joint fine-tuning allows the model to maintain its text retrieval capability (row 3 vs 1), though it results in a drop of over 2 points in multimodal retrieval accuracy (row 3 vs 2). In contrast, explicitly fine-tuning $M^{\text{hard}}$(NVEmb) for an addition 2K steps significantly boosts

Table 6: Ablation study to enhance model's text-to-text retrieval capability.

| Initialization | Training data | | M-BEIR∗ | BEIR∗ |
|---|---|---|---|---|
| | Multimodal | Text-to-Text | | |
| NVEmb | - | - | - | 62.9 |
| | ✓ | ✗ | 54.3 | 51.7 |
| | ✓ | ✓ | 52.2 | 63.0 |
| $M^{\text{hard}}$ (NVEmb) | - | - | 56.4 | 51.7 |
| | ✓ | ✓ | 55.6 | 63.1 |

∗ For M-BEIR, we only evaluate on the tasks with single-modality queries (i.e., tasks 1–5) while for BIER, we evaluate on 7 tasks: ArguAna, FiQA, NFCorpus, Quora, SCIDOCS, SciFact and TREC-COVID.

its text-to-text retrieval capability with a slight drop of 0.8 points in multimodal retrieval (row 5 vs 4).[12] This experiment shows that continuously fine-tuning a multimodal retriever to enhance its text-to-text retrieval is more effective than fine-tuning a retriever across all retrieval tasks simultaneously. This finding suggests that a better optimized curriculum learning strategy (Bengio et al., 2009) could further improve performance in universal multimodal retrieval, a direction we leave for future work.

### 5.3.3 STUDY ON PROMPTING MLLMS FOR RERANKING

In this section, we study the reranking effectiveness of MLLMs on all the tasks in the M-BEIR dataset. Specifically, for each development query, we rerank the top-10 retrieved candidates from $M^{\text{rand}}$(CLIP$_{\text{SF}}$). As shown in Table 7, prompting LLaVa-Next for reranking further boosts the rank-

---

[11]We follow Jiang et al. (2024) to prompt LLaVa-Next to output one word embedding for each query and document, i.e., <txt>\nSummary above sentence in one word:; \nSummary above image in one word:.

[12]Note that MM-Embed in Table 1 is fine-tuned under the same conditions for a total of 4.5K steps.

Table 7: Reranking study on top-10 retrieved candidates from $M^{\text{rand}}(\text{CLIP}_{\text{SF}})$ on the M-BEIR development query set.

| Task | Dataset | Retrieval | Rerank | |
|---|---|---|---|---|
| | | | 7B | 34B |
| 1. $q^{\text{txt}} \to c^{\text{img}}$ | VisualNews | 44.2 | 38.8 | 42.5 |
| | MSCOCO | 72.0 | 68.0 | 69.7 |
| | Fashion200K | 17.8 | 14.7 | 15.6 |
| 2. $q^{\text{txt}} \to c^{\text{txt}}$ | WebQA | 78.2 | 79.2 | 82.9 |
| 3. $q^{\text{txt}} \to (c^{\text{img}}, c^{\text{txt}})$ | EDIS | 48.3 | 46.5 | 47.4 |
| | WebQA | 78.2 | 67.7 | 68.3 |
| 4. $q^{\text{img}} \to c^{\text{txt}}$ | VisualNews | 37.4 | 29.3 | 29.8 |
| | MSCOCO | 91.0 | 87.3 | 89.0 |
| | Fashion200K | 17.3 | 9.9 | 12.0 |
| 5. $q^{\text{img}} \to c^{\text{img}}$ | NIGHTS | 32.1 | 29.4 | 32.7 |
| 6. $(q^{\text{img}}, q^{\text{txt}}) \to c^{\text{txt}}$ | OVEN | 40.6 | 43.2 | 43.7 |
| | InfoSeek | 25.6 | 28.4 | 29.0 |
| 7. $(q^{\text{img}}, q^{\text{txt}}) \to c^{\text{img}}$ | FashionIQ | 32.5 | 21.5 | 23.4 |
| | CIRR | 52.4 | 54.1 | 54.2 |
| 8. $(q^{\text{img}}, q^{\text{txt}}) \to (c^{\text{img}}, c^{\text{txt}})$ | OVEN | 60.6 | 63.8 | 63.7 |
| | InfoSeek | 45.3 | 48.7 | 50.5 |

ing accuracy in tasks 6-8, which involve multimodal queries (except for FashionIQ). However, the reranking degrades accuracy in tasks 1-5 which involve single-modal queries (except for WebQA task 2). This trend persists even after scaling the reranker from 7B to 34B (col 3, 2 vs 1).[13] We hypothesize that MLLM rerankers, as a more robust cross-encoder compared to a bi-encoder retriever, excel at challenging tasks involving multimodal queries, even in a zero-shot manner. However, zero-shot rerankers fail to leverage task- or domain-specific knowledge, which limits their performance on relatively simple tasks involving single-modal queries. The relevance signals between queries and documents in the News, Miscellaneous, and Fashion domains can vary significantly. Thus, optimizing prompts or instruction-tuning for MLLMs to better capture domain- or task-specific knowledge offers a promising direction for improving reranking accuracy.

## 6 CONCLUSION AND FUTURE WORK

In this paper, we present techniques for advancing information retrieval using multimodal large language models (MLLMs). We first study fine-tuning MLLM-based retrievers to tackle a general information retrieval scenario: universal multimodal retrieval, where models are required to deal with diverse retrieval tasks, multimodal queries, and documents. Our study shows that MLLM-based retrievers exhibit a *modality bias* in cross-modal retrieval tasks compared to CLIP-based retrievers. To address the issue, we propose modality-aware hard negative mining, which significantly improves our MLLM-based retrievers' accuracies by 5 points in the M-BEIR dataset, a benchmark for universal multimodal retrieval. Additionally, with our proposed continuous fine-tuning, our MLLM-based retriever, MM-Embed, is the first model to yield state-of-the-art retrieval accuracy in universal multimodal retrieval tasks while maintaining strong text-to-text retrieval capability (ranked top-5 on the MTEB retrieval tasks leaderboard). Finally, we explore prompting MLLMs as rerankers in M-BEIR tasks. We find that MLLMs can be used as zero-shot rerankers to further boost retrieval accuracy in challenging tasks that require the understanding of multimodal queries, such as visual question answering and composed image retrieval. For example, our zero-shot MLLM-based reranker improves the retrieval accuracy over the state-of-the-art retrievers by over 7 points in CIRCO.

Our work also suggests two promising future directions: (1) Distilling our MLLM-based retriever, MM-Embed, into smaller multimodal retrievers, such as CLIP (Radford et al., 2021) or BLIP (Li et al., 2022), to achieve better retrieval efficiency (see the efficiency comparisons in Appendix A.2); (2) Distilling an MLLM-based reranker into a retriever to further improve its retrieval capability in tasks involving multimodal queries. Other directions, such as iterative retrieval with relevance feedback (Han et al., 2024) and generative retrieval (Qu et al., 2024), are also worth exploring for universal multimodal retrieval tasks. In addition, recent work (Ma et al., 2024a; Faysse et al., 2024) has demonstrated that MLLMs can be fine-tuned to tackle visual document retrieval tasks, which could be incorporated into universal multimodal retrieval tasks.

---

[13]We use `llava-hf/llava-v1.6-34b-hf` in the experiment.

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

# A  APPENDIX

## A.1  IMPLEMENTATION DETAILS

We implement our training and inference using Tevatron (Gao et al., 2023). For CLIP-based retrievers, we follow all the settings from Wei et al. (2023). For the MLLM-based retrievers, we fine-tune models with DeepSpeed Zero 2 (Rajbhandari et al., 2020) and gradient checkpointing. During fine-tuning on the M-BEIR training data, we set the maximum length for queries and documents to 128. During continuously fine-tuning on both M-BEIR and text-to-text retrieval training data, we set the maximum length for queries and documents to 128 and 512, respectively. All fine-tuning is conducted on $8\times80$GB A100 GPUs. Note that image input only occupies a single token in length after being tokenized; however, each image will be converted to multiple image tokens. Thus, the actual input length for the MLLMs is longer than the maximum length we set. To speed up fine-tuning and inference for MLLM-based retrievers, we only use the global image patches, which occupy 576 ($24\times24$) image tokens.

## A.2  RETRIEVAL EFFICIENCY COMPARISONS

Table 8: Retrieval efficiency comparisons on M-BEIR dataset.

| Retriever | Storage (GBs) Index | Latency (ms) Encoding ($1^{st}$ / $50^{th}$ / $99^{th}$ perc.) | Vector search |
|---|---|---|---|
| $CLIP_{SF}$ | 16 | 26 / 27 / 39 | 6 |
| $BLIP_{FF}$ | 16 | 37 / 38 / 44 | 6 |
| MM-Embed | 86 | 81 / 194 / 203 | 33 |

Table 8 compares the retrieval efficiency in terms of storage and latency for different retrievers adopted in the paper. We measure the index storage required for the 5.6M documents from the M-BEIR dataset. As for retrieval latency, we measure the latencies of query encoding and vector search. For query latency, we randomly sample 100 queries from each test query pool in the 16 M-BEIR tasks and measure the per query encoding and vector search latency with a batch size of 1. Since query encoding latency varies with query length, we report the latency at $1^{th}$, $50^{th}$ and $99^{th}$ percentiles. The latency is measured using one thread on a Linux machine with a 2.2 GHz Intel Xeon Silver 4210 CPU and NVIDIA RTX A6000 GPUs, respectively. Note that we perform a brute-force search on the sharded index with two GPUs since the full index from UniEmb cannot be loaded into a single A6000 GPU.

## A.3  BASELINE REPRODUCING

Since we implement our fine-tuning and inference following the settings from Wei et al. (2023), our fine-tuned $M^{rand}(CLIP_{SF})$ should be equivalent to $CLIP_{SF}$ from Wei et al. (2023). In Table 9, we compare the results of our fine-tuned $M^{rand}(CLIP_{SF})$ with the checkpoint provided by the authors.[14]

Table 9: A comparison of $M^{rand}(CLIP_{SF})$ fine-tuned by us and Wei et al. (2023).

| Task | Dataset | $M^{rand}(CLIP_{SF})$ Wei et al. (2023) | Ours |
|---|---|---|---|
| M-BEIR Avg. | All | 47.4 | 47.4 |
| | Single-modal Qry | 52.5 | 51.7 |
| | multi-modal Qry | 39.1 | 40.1 |

---

[14]https://huggingface.co/TIGER-Lab/UniIR/blob/main/checkpoint/CLIP_SF/clip_sf_large.pth

Table 10: M-BEIR dataset statistics.

| Task | Dataset | Domain | # Query | | | # Relevance / Query | | | # Candid. |
|------|---------|--------|---------|-----|------|---------------------|-----|------|-----------|
| | | | Train | Dev | Test | Train | Dev | Test | |
| 1. $q^{txt} \rightarrow c^{img}$ | VisualNews (Liu et al., 2021a) | News | 99K | 20K | 20K | 1.0 | 1.0 | 1.0 | 542K |
| | MSCOCO (Lin et al., 2014) | Misc. | 100K | 24.8K | 24.8K | 1.0 | 1.0 | 1.0 | 5K |
| | Fashion200K (Han et al., 2017) | Fashion | 15K | 1.7K | 1.7K | 3.3 | 3.1 | 2.8 | 201K |
| 2. $q^{txt} \rightarrow c^{txt}$ | WebQA (Chang et al., 2022) | Wiki | 16K | 1.7K | 2.4K | 2.0 | 2.0 | 2.0 | 544K |
| 3. $q^{txt} \rightarrow (c^{img}, c^{txt})$ | EDIS (Liu et al., 2023b) | News | 26K | 3.2K | 3.2K | 2.6 | 2.6 | 2.6 | 1M |
| | WebQA (Chang et al., 2022) | Wiki | 16K | 1.7K | 2.4K | 1.4 | 1.4 | 1.4 | 544K |
| 4. $q^{img} \rightarrow c^{txt}$ | VisualNews (Liu et al., 2021a) | News | 100K | 20K | 20K | 1.0 | 1.0 | 1.0 | 537K |
| | MSCOCO (Lin et al., 2014) | Misc. | 113K | 5K | 5K | 5.0 | 5.0 | 5.0 | 25K |
| | Fashion200K (Han et al., 2017) | Fashion | 15K | 4.8K | 4.8K | 1.0 | 1.0 | 1.0 | 61K |
| 5. $q^{img} \rightarrow c^{img}$ | NIGHTS (Fu et al., 2023) | Misc. | 16K | 2K | 2K | 1.0 | 1.0 | 1.0 | 40K |
| 6. $(q^{img}, q^{txt}) \rightarrow c^{txt}$ | OVEN (Hu et al., 2023) | Wiki | 150K | 50K | 50K | 8.5 | 10.0 | 9.9 | 676K |
| | InfoSeek (Chen et al., 2023) | Wiki | 141K | 11K | 11K | 6.8 | 6.7 | 6.5 | 611K |
| 7. $(q^{img}, q^{txt}) \rightarrow c^{img}$ | FashionIQ (Wu et al., 2021) | Fashion | 16K | 2K | 6K | 1.0 | 1.0 | 1.0 | 74K |
| | CIRR (Liu et al., 2021b) | Misc. | 26K | 2K | 4K | 1.0 | 1.0 | 1.0 | 21K |
| 8. $(q^{img}, q^{txt}) \rightarrow (c^{img}, c^{txt})$ | OVEN (Hu et al., 2023) | Wiki | 157K | 14.7K | 14.7K | 17.8 | 17.5 | 17.7 | 335K |
| | InfoSeek (Chen et al., 2023) | Wiki | 143K | 17.6K | 17.6K | 9.1 | 7.5 | 7.5 | 481K |
| M-BEIR (Wei et al., 2023) | | 4 domains | 1.1M | 182K | 190K | 6.5 | 5.9 | 5.7 | 5.6M |

Table 11: Document modality statistics of positive samples in M-BEIR train set. We observe that there are more text-modal samples in the positive samples of the M-BIER training set. The statistics indicate that text modality has a higher probability of being randomly sampled as negatives than other modalities (i.e., image and interleaved text-image); thus, the text modality bias of MLLM-based retrievers is not from the sampling bias of the random sampling strategy.

| Document modality | Text | Image | Interleaved text-image |
|-------------------|------|-------|------------------------|
| number | 568,757 | 364,903 | 399,130 |
| percentage | 42.6% | 27.4% | 29.9% |

Table 12: A comparison with other existing state-of-the-art multimodal retrieval models on M-BEIR dataset.

| Task | Dataset | E5-V (Jiang et al., 2024) | MagicLens (Zhang et al., 2024) | MM-Embed |
|------|---------|---------------------------|-------------------------------|----------|
| M-BEIR Avg. | All | 11.5 | 5.8 | 52.7 |
| | Single-modal Qry | 14.6 | 8.1 | 56.1 |
| | Multi-modal Qry | 6.3 | 2.0 | 47.0 |

Table 13: A comparison with other existing state-of-the-art multimodal retrieval models on M-BEIR subtasks in a local evaluation setting (Wei et al., 2023), where retrieval is conducted only among the corpus of each dataset consisting of less than 1M candidate documents with a single modality.

| Task | Dataset | E5-V (Jiang et al., 2024) | MagicLens (Zhang et al., 2024) | MM-Embed |
|------|---------|---------------------------|-------------------------------|----------|
| 1. $q^{txt} \rightarrow c^{img}$ | MSCOCO | 75.8 | 68.5 | 82.7 |
| 2. $q^{txt} \rightarrow c^{txt}$ | WebQA | 84.8 | 47.9 | 96.6 |
| 4. $q^{img} \rightarrow c^{txt}$ | MSCOCO | 83.4 | 17.4 | 91.0 |
| 5. $q^{img} \rightarrow c^{img}$ | NIGHTS | 26.7 | 14.1 | 32.7 |
| 7. $(q^{img}, q^{txt}) \rightarrow c^{img}$ | FashionIQ | 8.4 | 13.8 | 26.0 |
| | CIRR | 30.1 | 37.5 | 53.0 |

Table 14: Retrieval models' top-1 modality accuracy (M.A.@1). We observe that most MLLM-based retrievers suffer from low modality accuracy on Task 1 due to modality bias, particularly in NVEmb due to its superior text retrieval capability. This issue can be resolved with our modality-aware hard negative mining. Even though MM-Embed exhibits strong text retrieval effectiveness, no modality bias is observed.

| Task | Dataset | $M^{\text{rand}}$ | | | | $M^{\text{hard}}$ | | | MM-Embed |
|------|---------|------|------|------|------|------|------|------|----------|
| | | CLIP$_{\text{SF}}$ | LLaVa-E | LLaVa-P | NVEmb | CLIP$_{\text{SF}}$ | LLaVa-P | NVEmb | |
| 1. $q^{\text{txt}} \rightarrow c^{\text{img}}$ | VisualNews | 0.97 | 0.82 | 0.94 | 0.76 | 0.98 | 1.00 | 1.00 | 1.00 |
| | MSCOCO | 0.93 | 0.80 | 0.91 | 0.42 | 0.98 | 0.99 | 1.00 | 1.00 |
| | Fashion200K | 0.97 | 1.00 | 1.00 | 0.78 | 1.00 | 1.00 | 1.00 | 0.99 |
| 2. $q^{\text{txt}} \rightarrow c^{\text{txt}}$ | WebQA | 1.00 | 1.00 | 1.00 | 1.00 | 1.00 | 1.00 | 0.99 | 1.00 |
| 3. $q^{\text{txt}} \rightarrow (c^{\text{img}}, c^{\text{txt}})$ | EDIS | 0.94 | 1.00 | 0.96 | 1.00 | 0.90 | 0.99 | 1.00 | 0.99 |
| | WebQA | 1.00 | 1.00 | 1.00 | 1.00 | 1.00 | 1.00 | 1.00 | 1.00 |
| 4. $q^{\text{img}} \rightarrow c^{\text{txt}}$ | VisualNews | 0.33 | 0.90 | 0.84 | 0.84 | 0.97 | 1.00 | 1.00 | 1.00 |
| | MSCOCO | 0.99 | 0.99 | 1.00 | 0.96 | 0.94 | 0.99 | 1.00 | 0.99 |
| | Fashion200K | 0.99 | 0.99 | 1.00 | 0.99 | 1.00 | 1.00 | 1.00 | 1.00 |
| 5. $q^{\text{img}} \rightarrow c^{\text{img}}$ | NIGHTS | 1.00 | 1.00 | 1.00 | 1.00 | 1.00 | 1.00 | 0.99 | 1.00 |
| 6. $(q^{\text{img}}, q^{\text{txt}}) \rightarrow c^{\text{txt}}$ | OVEN | 1.00 | 1.00 | 1.00 | 1.00 | 1.00 | 1.00 | 1.00 | 0.99 |
| | InfoSeek | 0.94 | 1.00 | 1.00 | 1.00 | 0.97 | 0.99 | 1.00 | 0.99 |
| 7. $(q^{\text{img}}, q^{\text{txt}}) \rightarrow c^{\text{img}}$ | FashionIQ | 0.99 | 1.00 | 1.00 | 1.00 | 1.00 | 1.00 | 1.00 | 1.00 |
| | CIRR | 0.99 | 1.00 | 1.00 | 1.00 | 1.00 | 1.00 | 1.00 | 1.00 |
| 8. $(q^{\text{img}}, q^{\text{txt}}) \rightarrow (c^{\text{img}}, c^{\text{txt}})$ | OVEN | 1.00 | 1.00 | 1.00 | 1.00 | 1.00 | 1.00 | 1.00 | 1.00 |
| | InfoSeek | 0.98 | 1.00 | 1.00 | 1.00 | 1.00 | 1.00 | 1.00 | 1.00 |

Table 15: Detailed results on MTEB retrieval tasks.

| Model | AA | CF | CQ | DB | Fe | FQ | HQ | MS | NF | NQ | Qu | SD | SF | T2 | TC | Avg. |
|-------|------|------|------|------|------|------|------|------|------|------|------|------|------|------|------|------|
| NVEmb (Lee et al., 2024) | 68.2 | 34.7 | 50.5 | 48.3 | 87.8 | 63.1 | 79.9 | 46.5 | 38.0 | 71.2 | 89.2 | 20.2 | 78.4 | 28.4 | 85.9 | 59.4 |
| $M^{\text{Rand}}$ (LLaVa-P) | 48.4 | 12.9 | 34.0 | 34.0 | 52.2 | 33.7 | 50.1 | 12.3 | 30.4 | 36.5 | 83.8 | 17.9 | 72.3 | 73.4 | 19.6 | 40.8 |
| $M^{\text{Rand}}$ (NVEmb) | 51.5 | 23.7 | 43.6 | 44.9 | 78.6 | 46.5 | 70.2 | 32.5 | 38.9 | 54.1 | 87.5 | 20.3 | 74.5 | 83.4 | 23.4 | 51.6 |
| $M^{\text{hard}}$ (LLaVa-P) | 38.6 | 20.4 | 38.0 | 36.9 | 78.1 | 36.2 | 61.2 | 23.2 | 35.1 | 45.1 | 86.1 | 19.2 | 72.7 | 27.7 | 77.2 | 46.4 |
| $M^{\text{hard}}$ (NVEmb) | 37.2 | 30.8 | 44.0 | 44.3 | 86.4 | 45.5 | 70.6 | 34.2 | 37.4 | 49.7 | 86.9 | 13.9 | 64.1 | 23.5 | 76.7 | 49.7 |
| MM-Embed | 69.0 | 39.3 | 49.7 | 50.6 | 92.6 | 60.1 | 81.4 | 45.1 | 40.5 | 70.6 | 88.7 | 21.8 | 78.3 | 31.1 | 85.4 | 60.3 |

* Dataset Legend: AA=ArguAna, CF=Climate-FEVER, CQ=CQADupStack, DB=DBPedia, Fe=FEVER, FQ=FiQA, HQ=HotpotQA, MS=MSMARCO, NF=NFCorpus, NQ=NaturalQuestions, Qu=Quora, SD=SCIDOCS, SF=SciFact, T2=Touché-2020, TC=TREC-COVID

Table 16: NVEmb (and LLaVa-E) instructions for M-BEIR and MTEB, which are from Wei et al. (2023) and Lee et al. (2024), respectively. For all the candidates, we use the prompt to generate the embedding: $< c^{\text{img}} > \backslash n < c^{\text{txt}} > <eos>$.

| Task | Dataset | M-BEIR task instruction |
|------|---------|-------------------------|
| 1. $q^{\text{txt}} \rightarrow c^{\text{img}}$ | VisualNews | *Identify the news-related image in line with the described event.\nQuery: $< q^{\text{txt}} > <eos>$* |
| | MSCOCO | *Find me an everyday image that matches the given caption.\nQuery: $< q^{\text{txt}} > <eos>$* |
| | Fashion200K | *Based on the following fashion description, retrieve the best matching image.\nQuery: $< q^{\text{txt}} > <eos>$* |
| 2. $q^{\text{txt}} \rightarrow c^{\text{txt}}$ | WebQA | *Retrieve passages from Wikipedia that provide answers to the following question.\nQuery: $< q^{\text{txt}} > <eos>$* |
| 3. $q^{\text{txt}} \rightarrow (c^{\text{img}}, c^{\text{txt}})$ | EDIS | *Find a news image that matches the provided caption.\nQuery: $< q^{\text{txt}} > <eos>$* |
| | WebQA | *Find a Wikipedia image that answers this question.\nQuery: $< q^{\text{txt}} > <eos>$* |
| 4. $q^{\text{img}} \rightarrow c^{\text{txt}}$ | VisualNews | *Find a caption for the news in the given photo.\nQuery: $< q^{\text{img}} > <eos>$* |
| | MSCOCO | *Find an image caption describing the following everyday image.\nQuery: $< q^{\text{img}} > <eos>$* |
| | Fashion200K | *Find a product description for the fashion item in the image.\nQuery: $< q^{\text{img}} > <eos>$* |
| 5. $q^{\text{img}} \rightarrow c^{\text{img}}$ | NIGHTS | *Find a day-to-day image that looks similar to the provided image.\nQuery: $< q^{\text{img}} > <eos>$* |
| 6. $(q^{\text{img}}, q^{\text{txt}}) \rightarrow c^{\text{txt}}$ | OVEN | *Retrieve a Wikipedia paragraph that provides an answer to the given query about the image.\nQuery: $< q^{\text{img}} > \backslash n < q^{\text{img}} > <eos>$* |
| | InfoSeek | *Retrieve a Wikipedia paragraph that provides an answer to the given query about the image.\nQuery: $< q^{\text{img}} > \backslash n < q^{\text{img}} > <eos>$* |
| 7. $(q^{\text{img}}, q^{\text{txt}}) \rightarrow c^{\text{img}}$ | FashionIQ | *Find a fashion image that aligns with the reference image and style note.\nQuery: $< q^{\text{img}} > \backslash n < q^{\text{img}} > <eos>$* |
| | CIRR | *Retrieve a day-to-day image that aligns with the modification instructions of the provided image.\nQuery: $< q^{\text{img}} > \backslash n < q^{\text{img}} > <eos>$* |
| 8. $(q^{\text{img}}, q^{\text{txt}}) \rightarrow (c^{\text{img}}, c^{\text{txt}})$ | OVEN | *Retrieve a Wikipedia image-description pair that provides evidence for the question of this image.\nQuery: $< q^{\text{img}} > \backslash n < q^{\text{img}} > <eos>$* |
| | InfoSeek | *Retrieve a Wikipedia image-description pair that provides evidence for the question of this image.\nQuery: $< q^{\text{img}} > \backslash n < q^{\text{img}} > <eos>$* |

| Task | Dataset | MTEB task instruction |
|------|---------|------------------------|
| 9. $q^{\text{txt}} \rightarrow c^{\text{txt}}$ | ArguAna | *Given a claim, find documents that refute the claim\nQuery: $< q^{\text{txt}} > <eos>$* |
| | Climate-FEVER | *Given a claim about climate change, retrieve documents that support or refute the claim\nQuery: $< q^{\text{txt}} > <eos>$* |
| | CQADupStack | Given a question, retrieve detailed question descriptions from Stackexchange that are duplicates to the given question\nQuery: $< q^{\text{txt}} > <eos>$ |
| | DBPedia | *Given a query, retrieve relevant entity descriptions from DBPedia\nQuery: $< q^{\text{txt}} > <eos>$* |
| | FEVER | Given a claim, retrieve documents that support or refute the claim\nQuery: $< q^{\text{txt}} > <eos>$ |
| | FiQA | *Given a financial question, retrieve user replies that best answer the question\nQuery: $< q^{\text{txt}} > <eos>$* |
| | HotpotQA | *Given a multi-hop question, retrieve documents that can help answer the question\nQuery: $< q^{\text{txt}} > <eos>$* |
| | MSMARCO | *Given a web search query, retrieve relevant passages that answer the query\nQuery: $< q^{\text{txt}} > <eos>$* |
| | NFCorpus | *Given a question, retrieve relevant documents that best answer the question\nQuery: $< q^{\text{txt}} > <eos>$* |
| | NaturalQuestions | *Given a question, retrieve Wikipedia passages that answer the question\nQuery: $< q^{\text{txt}} > <eos>$* |
| | Quora | *Find questions that have the same meaning as the input question\nQuery: $< q^{\text{txt}} > <eos>$* |
| | SICDOCS | *Given a scientific paper title, retrieve paper abstracts that are cited by the given paper\nQuery: $< q^{\text{txt}} > <eos>$* |
| | SciFact | *Given a scientific claim, retrieve documents that support or refute the claim\nQuery: $< q^{\text{txt}} > <eos>$* |
| | Touch´e-2020 | *Given a question, retrieve detailed and persuasive arguments that answer the question\nQuery: $< q^{\text{txt}} > <eos>$* |
| | TREC-COVID | *Given a query on COVID-19, retrieve documents that answer the query\nQuery: $< q^{\text{txt}} > <eos>$* |

Table 17: LLaVa-P instructions for M-BEIR and MTEB. [image], [text] and [image,text] are used to inform LLaVa-P the user desired modality. For all the candidates, we use the prompt to generate the embedding: $< c^{img} >\backslash n< c^{txt} >\backslash nDescribe~the~above~in~one~word:$

| Task | Dataset | M-BEIR task instruction |
|---|---|---|
| 1. $q^{txt} \rightarrow c^{img}$ | VisualNews | *[image] $< q^{txt} >\backslash nDescribe the news-related caption in one word:* |
| | MSCOCO | *[image] $< q^{txt} >\backslash nDescribe the everyday caption in one word:* |
| | Fashion200K | *[image] $< q^{txt} >\backslash nDescribe the fashion description in one word:* |
| 2. $q^{txt} \rightarrow c^{txt}$ | WebQA | *[text] $< q^{txt} >\backslash nAnswer the question using Wikipedia in one word:* |
| 3. $q^{txt} \rightarrow (c^{img}, c^{txt})$ | EDIS | *[image,text] $< q^{txt} >\backslash nDescribe the news-related caption in one word:* |
| | WebQA | *[image,text] $< q^{txt} >\backslash nAnswer the question using Wikipedia in one word:* |
| 4. $q^{img} \rightarrow c^{txt}$ | VisualNews | *[text] $< q^{img} >\backslash nDescribe the news-related image in one word:* |
| | MSCOCO | *[text] $< q^{img} >\backslash nDescribe the everyday image in one word:* |
| | Fashion200K | *[text] $< q^{img} >\backslash nDescribe the fashion image in one word:* |
| 5. $q^{img} \rightarrow c^{img}$ | NIGHTS | *[image] $< q^{img} >\backslash nDescribe the everyday image in one word:* |
| 6. $(q^{img}, q^{txt}) \rightarrow c^{txt}$ | OVEN InfoSeek | [text] $< q^{img} >\backslash n< q^{txt} >\backslash nAnswer the question based on the image from Wikipedia in one word:* |
| 7. $(q^{img}, q^{txt}) \rightarrow c^{img}$ | FashionIQ | *[image] $< q^{img} >\backslash nChange the style of this shirt/dress/toptee to $< q^{txt} >\backslash nDescribe this modified shirt/dress/toptee in one word:* |
| | CIRR | *[image] $< q^{img} >\backslash nModify this image with $< q^{txt} >\backslash nDesribe modified image in one word:* |
| 8. $(q^{img}, q^{txt}) \rightarrow (c^{img}, c^{txt})$ | OVEN InfoSeek | *[image,text] $< q^{img} >\backslash n< q^{txt} >\backslash nAnswer the question based on the interleaved image-text passage from Wikipedia in one word:* |

| Task | Dataset | MTEB task instruction |
|---|---|---|
| 9. $q^{txt} \rightarrow c^{txt}$ | ArguAna | *[text] $< q^{txt} >\backslash nGiven a claim, generate a document that refute the claim in one word:* |
| | Climate-FEVER | *[text] $< q^{txt} >\backslash nGiven a claim about climate change, generate a document that supports or refutes the claim in one word:* |
| | CQADupStack | *[text] $< q^{txt} >\backslash nDescribe the Stackexchange question in one word:* |
| | DBPedia | *[text] $< q^{txt} >\backslash nGiven a query, generate a relevant entity description from DBPedia in one word:* |
| | FEVER | *[text] $< q^{txt} >\backslash nGiven a claim, generate a document that supports or refutes the claim in one word:* |
| | FiQA | *[text] $< q^{txt} >\backslash nAnswer the financial question in one word:* |
| | HotpotQA | *[text] $< q^{txt} >\backslash nAnswer the multi-hop question in one word:* |
| | MSMARCO | *[text] $< q^{txt} >\backslash nAnswer the web search query in one word:* |
| | NFCorpus | *[text] $< q^{txt} >\backslash nAnswer the question in one word:* |
| | NaturalQuestions | *[text] $< q^{txt} >\backslash nAnswer the question using Wikipedia in one word:* |
| | Quora | *[text] $< q^{txt} >\backslash nDescribe the question in one word:* |
| | SICDOCS | *[text] $< q^{txt} >\backslash nGiven a scientific paper title, generate a paper abstract that is cited by the given paper in one word:* |
| | SciFact | *[text] $< q^{txt} >\backslash nGiven a scientific claim, generate a document that support or refute the claim in one word:* |
| | Touch´e-2020 | *[text] $< q^{txt} >\backslash nAnswer the question with detailed and persuasive arguments in one word:* |
| | TREC-COVID | *[text] $< q^{txt} >\backslash nAnswer the query on COVID-19 in one word:* |

Table 18: Prompts for reranking tasks in M-BEIR .

| Task | Dataset | Prompt |
|---|---|---|
| 1. $q^{txt} \rightarrow c^{img}$ | VisualNews | *$< c^{img} >\backslash nNews:< q^{txt} >\backslash nDoes the above News image match the News story? True or False* |
| | MSCOCO | *$< c^{img} >\backslash nCaption:< q^{txt} >\backslash nDoes the above daily-life image match the caption? True or False* |
| | Fashion200K | *$< c^{img} >\backslash nDescription:< q^{txt} >\backslash nDoes the above image match the cloth style description? True or False* |
| 2. $q^{txt} \rightarrow c^{txt}$ | WebQA | *Question: $< q^{txt} >\backslash nAnswer: < c^{txt} >\backslash nDoes the answer correctly answer the question? True or False* |
| 3. $q^{txt} \rightarrow (c^{img}, c^{txt})$ | EDIS WebQA | *Question: $< q^{txt} >\backslash nAnswer: < c^{txt} >\backslash nDoes the answer correctly answer the question? True or False* |
| 4. $q^{img} \rightarrow c^{txt}$ | VisualNews | *$< q^{img} >\backslash nNews:< c^{txt} >\backslash nDoes the above News image match the News story? True or False* |
| | MSCOCO | *$< q^{img} >\backslash nCaption:< c^{txt} >\backslash nDoes the above daily-life image match the caption? True or False* |
| | Fashion200K | *$< q^{img} >\backslash nDescription:< c^{txt} >\backslash nDoes the above image match the cloth style description? True or False* |
| 5. $q^{img} \rightarrow c^{img}$ | NIGHTS | *$< q^{img} >\backslash n< c^{img} >\backslash nDoes the above two images have the same scene? True or False* |
| 6. $(q^{img}, q^{txt}) \rightarrow c^{txt}$ | OVEN InfoSeek | *$< q^{img} >\backslash nQuestion:< q^{txt} >\backslash nAnswer:< c^{txt} >Does the answer correctly answer the question? True or False* |
| 7. $(q^{img}, q^{txt}) \rightarrow c^{img}$ | FashionIQ CIRR | *$< c^{img} >\backslash nCaption:< q^{txt} >\backslash nDoes the above caption describe the modification of the image? True or False* |
| 8. $(q^{img}, q^{txt}) \rightarrow (c^{img}, c^{txt})$ | OVEN InfoSeek | *$< q^{img} >\backslash nQuestion:< q^{txt} >\backslash nAnswer:< c^{txt} >Does the answer correctly answer the question? True or False* |

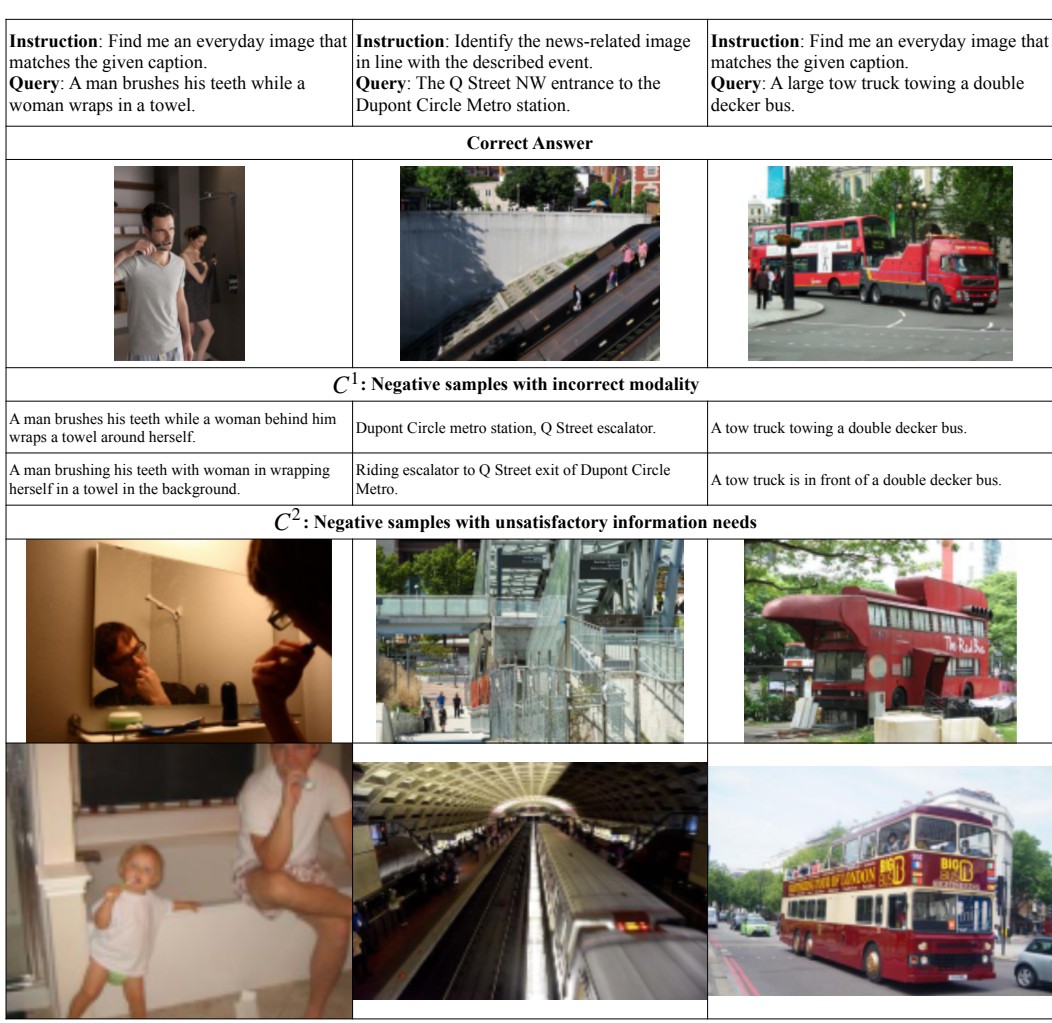

Figure 2: Examples of modality-aware negative samples mined by $M^{\text{rand}}$(NVEmb). We observe that negative samples with incorrect modality show similar semantic meaning to queries, while negative samples with unsatisfactory information needs provide less accurate information compared to the correct answers.

| M-BEIR CIRR Task 7 | | | |
|---|---|---|---|
| Query | Answer | Retrieval | Reranking |
| Human and one animal from a different specie |  |  |  |
| Same breed dog, focus on its head. |  |  |  |
| Put the fries in a white plate with white background, clean. |  |  |  |
| M-BEIR FashionIQ Task 7 | | | |
| Query | Answer | Retrieval | Reranking |
| Is shiny and silver with shorter sleeves and fit and flare. |  |  |  |
| Is grey with black design and is a light printed short dress. |  |  |  |
| Is a solid red color and shorter and tighter with more blue and white. |  |  |  |

Figure 3: Top-1 candidates for the tasks of composed image retrieval and reranking. In many cases, retrieval and reranking yield different top-1 results from labeled positives, but these results appear to be correct since each query only has a single labeled positive candidate (see Table 10).

