# OpenReview forum: "MM-EMBED: UNIVERSAL MULTIMODAL RETRIEVAL WITH MULTIMODAL LLMS"
_ICLR.cc/2025/Conference — ICLR 2025 Poster_

### Official Review · Reviewer_8yb7 · 2024-10-29

**Soundness:** 3
**Presentation:** 3
**Contribution:** 2
**Rating:** 8
**Confidence:** 4

**Summary:**

This paper proposes a general retrieval method for multimodal retrieval tasks, named UniEmb, based on multimodal large language models. The paper identifies the phenomenon of modality bias in multimodal retrieval and finds that current retrieval methods based on multimodal large language are lacking in text retrieval performance. To address these issues, the authors propose a continued fine-tuning method based on hard negative samples, as well as a strategy that uses multimodal large language for further re-ranking. The proposed method is comprehensively evaluated on two widely adopted datasets, M-BEIR and MTEB, with results showing that UniEmb improves text retrieval performance while maintaining the effectiveness of multimodal retrieval.

**Strengths:**

1. The paper identifies an interesting issue of modality bias in multimodal retrieval based on multimodal large language models, and it proposes an effective hard sample mining strategy to address this issue while preserving text retrieval capabilities.
2. The paper introduces a zero-shot re-ranking approach using multimodal large language model, which, to some extent, improves multimodal retrieval accuracy.
3. The evaluation is thorough, using widely adopted M-BEIR and MTEB datasets, and the paper provides detailed parameters for reproducibility, lending credibility to the experimental results.

**Weaknesses:**

1. The proposed methods for hard sample mining and re-ranking using large multimodal models are straightforward and intuitive, and may lack some novelty.
2. There is no direct comparison with similar baseline models, such as E5-V, on the M-BEIR dataset.

**Questions:**

1. In Table 1, why is the M^rand column for MTEB Text Retrieval empty? Are these models not evaluable on MTEB? I believe that comparing M^rand with M^hard would more effectively demonstrate the modality bias issue and the effectiveness of the proposed method.
2. In Table 4, why does re-ranking show limited improvement for knowledge-related multimodal retrieval datasets like OVEN and InfoSeek? and even a decline in performance on some tasks (e.g., task1 and task3)? How do the authors explain this phenomenon, and could it be related to the modality of the retrieval task?
3. In line 343, the authors mention that MLLM-based retrievers tend to retrieve text over images. Could this be related to the sampling of negative samples during training? If there are more image samples in the dataset, this could lead to a higher proportion of images as negative samples, which could further induce bias. I suggest conducting a statistical analysis on the modality distribution of the data samples.

---

> ### Author Response · Authors · 2024-11-20
>
> **Re: (Weakness1) The proposed methods for hard sample mining and re-ranking using large multimodal models are straightforward and intuitive, and may lack some novelty.**
>
> We thank you for your comment. The main contribution of the work is that we are the first to build a universal multimodal retriever, which reaches state-of-the-art retrieval effectiveness on both multimodal and text-only retrieval while most of the existing work only focuses on one or just a few limited retrieval tasks. Additionally, it is an important finding that MLLMs can be prompted as a strong zero-shot reranker for those complex and challenging tasks involving multimodal queries since such retrieval training data is hard to collect. We believe this study and findings are novel and provide valuable insights to the community.
>
> **Re: (Weakness2) There is no direct comparison with similar baseline models, such as E5-V, on the M-BEIR dataset.**
>
> We thank you for the insightful questions and agree that we should make a comparison with the concurrent work. Please see the general response.
>
> **Re: (Q1) In Table 1, why is the M^rand column for MTEB Text Retrieval empty? Are these models not evaluable on MTEB? I believe that comparing M^rand with M^hard would more effectively demonstrate the modality bias issue and the effectiveness of the proposed method.**
>
> We thank you for the insightful suggestion. We have ran inference for $M^{\text{rand}}$(LLaVa-P) and $M^{\text{rand}}$(NVEmb) to compare with their $M^{\text{hard}}$ counterparts as shown below. We observe that $M^{\text{hard}}$(NVEmb) show slight degradation compared to $M^{\text{rand}}$(NVEmb), which is expected, since the hard negative mining aims to address the text retrieval modality bias in NVEmb. On the other hand, interestingly, a reverse trend can be observed in  LLaVa-P.  We hypothesize that $M^{\text{rand}}$(LLaVa-P) has no such worse text retrieval bias issue compared to $M^{\text{rand}}$(NVEmb); thus, still improve the general retrieval capability in hard negative mining training.
>
> |MTEB Text Retrieval |$M^{\text{rand}}$(LLaVa-P)  |$M^{\text{rand}}$(NVEmb) |$M^{\text{hard}}$(LLaVa-P)  |$M^{\text{hard}}$(NVEmb) |
> | ----------- | ------------- | ------ | ------ |------ |
> nDCG@10| 40.8| 51.6| 46.4| 49.7|
>
> **Re: (Q2) In Table 4, why does re-ranking show limited improvement for knowledge-related multimodal retrieval datasets like OVEN and InfoSeek? and even a decline in performance on some tasks (e.g., task1 and task3)? How do the authors explain this phenomenon, and could it be related to the modality of the retrieval task?**
>
> We thank you for the insightful questions. If we understand correctly, you want to ask why the zero-shot re-ranking does not show significant effectiveness improvement in Table 3 (task 6 and 7). We hypothesize that the fine-tuned MLLM retrievers also learn some task-specific knowledge compared to the zero-shot reranker. Thus, when observing the results in Table 4, we can see a significant gap between the retrieval and reranked results, where all fine-tuned retrievers are not specifically fine-tuned on the task of CIRCO. We will add the discussion and our hypothesis in the revised manuscript.
>
> **Re: (Q3)  In line 343, the authors mention that MLLM-based retrievers tend to retrieve text over images. Could this be related to the sampling of negative samples during training? If there are more image samples in the dataset, this could lead to a higher proportion of images as negative samples, which could further induce bias. I suggest conducting a statistical analysis on the modality distribution of the data samples.**
>
> We thank you for the insightful questions and agree that we should investigate the negative sampling distribution in our random negative sampling approach. The below table reports the total number of documents in text, image or text-image interleaved modalities in train set. We observe that while randomly sampling negatives, text is the most likely sampled modality but MLLM-based retrievers still tend to retrieve the documents with text modality. This means that text modality bias is not resulted from random negative sampling. This analysis provides additional evidence of text retrieval bias on MLLM-based retrievers. We thank you for the helpful suggestions and will add the analysis in our Appendix.
>
> ||Text  |Image |Interleaved Text-Image |
> | ----------- | ------------- | ------ | ------ |
> |Number| 568,757 | 364,903 | 399,130 |
> |Percentage| 42.6% | 27.4% | 29.9%|

---

> > ### Comment · Reviewer_8yb7 · 2024-11-26
> > **Official Comment by Reviewer 8yb7**
> >
> > I appreciate the authors' detailed response to my questions, which addressed most of my concerns. After reviewing the authors' replies to other reviews and the revised manuscript, I will raise my score to 8 due to the solid experiments and thorough discussion.

---

> > > ### Author Response · Authors · 2024-11-26
> > >
> > > We also appreciate your insightful comments and suggestions, which help improve our manuscript and research!

---

### Official Review · Reviewer_ESAe · 2024-10-30

**Soundness:** 3
**Presentation:** 2
**Contribution:** 2
**Rating:** 6
**Confidence:** 4

**Summary:**

The paper presents techniques for advancing information retrieval with multimodal large language models (MLLMs). It first finetunes MLLM-based retrievers to tackle universal multimodal retrieval tasks and shows that MLLM-based retrievers exhibit modality bias in cross-modal retrieval tasks compared to CLIP-based retrievers. To address the issue, it proposes modality-aware hard negative mining and continual fine-tuning, and the MLLM-based retriever, UniEmb, yields state-of-the-art retrieval accuracy in universal multimodal retrieval tasks while maintaining strong text-to-text retrieval capability. Moreover, it explores to prompt MLLMs as a re-ranker to further boost retrieval accuracy.

**Strengths:**

1.	The paper is well-motivated to use multimodal LLMs for advancing information retrieval and support broader retrieval scenarios.
2.	The experiments are relatively comprehensive to validate the effectiveness of the proposed method.

**Weaknesses:**

1.	The paper claims that existing retrieval models typically addressed search scenario where retrieval task are fixed and only a single modality is supported for both queries and retrieved results, but a series of retrieval models for multimodal retrieval are listed in related work.
2.	The paper claims that it is the first work to explore prompting MLLMs fir zero-shot reranking, but as far as I know, the work [a] has used MLLMs for reranking, and the differences should be explained in detail.
3.	Some expressions are confused. For example, in Line 308 of Page 6, “we initialized retriever using the pretrained model”, does the “retriever” refer to the model to be fine-tuned with hard negatives?
4.	It is not easy to understand Figure 1, where the task definition of Task 1-13 is missing. The relationship between the upper left and lower right sub-figure is not clearly illustrated.
5.	How is the performance when using fine-tuned MLLMs for reranking? Is it better than reranking in a zero-shot manner for tasks that involve single-modal queries?
6.	More qualitative cases are suggested to be provided to better illustrate the effects, such as how modality-aware hard negative mining alleviates modality bias.

[a] Qu L, Li H, Wang T, et al. Unified Text-to-Image Generation and Retrieval[J]. arXiv preprint arXiv:2406.05814, 2024.

**Questions:**

Please refer to the Weaknesses section and address my concerns.

---

> ### Author Response · Authors · 2024-11-20
>
> Many thanks for your comments and feedback. We discuss your raised points in the following.
>
> **Re: (Weakness1) The paper claims that existing retrieval models typically addressed search scenario where retrieval task are fixed and only a single modality is supported for both queries and retrieved results, but a series of retrieval models for multimodal retrieval are listed in related work.**
>
> We thank you for the comments. Although we list a series of multimodal retrieval work in related work for a comparison, most of the work only focuses on limited  retrieval scenarios. For example, Liu et al.[1] only explore multimodal retrieval in a single task, WebQA (which is one of the subtask in our evaluation). The more recent work[2][3] also only develops multimodal retrievers for limited tasks (e.g., MSCOCO and Flickr). Furthermore, all the work simply assumes that all the documents in the corpus are in the same modality while our work considers the most practical retrieval scenario, where documents with diverse modalities are included. We highlight and address the issue of modality bias, and build a SoTA universal retriever under such challenging scenarios. We will revise our manuscript to clarify the claim. We also provide a more comprehensive empirical comparison with the recent models in general response as another supporting evidence.
>
> [1] [Universal vision-language dense retrieval: Learning a unified representation space for multi-modal retrieval.](https://arxiv.org/pdf/2209.00179)
>
> [2] [Jina CLIP: Your CLIP Model Is Also Your Text Retriever](https://arxiv.org/abs/2405.20204)
>
> [3] [E5-V: Universal Embeddings with Multimodal Large Language Models](https://arxiv.org/pdf/2407.12580)
>
>
> **Re: (Weakness2) The paper claims that it is the first work to explore prompting MLLMs for zero-shot reranking, but as far as I know, the work [a] has used MLLMs for reranking, and the differences should be explained in detail.**
>
> We thank you for providing the valuable reference[1]. We agree that a comparison with the reference work should be made. There are two main differences between our approaches. First, the reference paper leverages query likelihood[2] (i.e., the probability of generating the text query given the image as answer) for reranking, which is only suitable when queries are in text format. In contrast, our prompt-based approach leverages MLLMs’ instruction following capability and formulates any multimodal retrieval tasks into instruction following tasks. Second, the reference paper is mainly to improve the zero-shot retrieval approach while we explore zero-shot reranking upon the strong retrievers which have been fine-tuned on diverse training data for multimodal retrieval. We have added the comparison in our Related work (see our Related Work in the revised manuscript) and also see the potential to extend the query likelihood approach to more diverse multimodal retrieval tasks (see lines 534--536 in the revised manuscript).
>
> [1] [Unified Text-to-Image Generation and Retrieval](https://arxiv.org/pdf/2406.05814)
>
> [2] [Open-source Large Language Models are Strong Zero-shot Query Likelihood Models for Document Ranking](https://aclanthology.org/2023.findings-emnlp.590/)
>
>
> **Re: (Weakness3) Some expressions are confused. For example, in Line 308 of Page 6, “we initialized retriever using the pretrained model”, does the “retriever” refer to the model to be fine-tuned with hard negatives?**
>
> Thank you for the careful examination and your understanding is correct. While fine-tuning MLLMs with hard negatives, we retrain the models (i.e., initialize with pre-trained MLLMs) rather than continuously fine-tuning the $M^{\text{rand}}$ models. We have revised our manuscript accordingly (see lines 308--309 in the revised manuscript).
>
>
> **Re: (Weakness4) It is not easy to understand Figure 1, where the task definition of Task 1-13 is missing. The relationship between the upper left and lower right sub-figure is not clearly illustrated.**
>
> We thank you for the helpful suggestion. We have revised our Figure 1 and caption to clarify the task definitions and the illustrations of retriever fine-tuning and zero-shot reranking (see Figure 1 in the revised manuscript).
>
> Continue...

---

> > ### Author Response · Authors · 2024-11-20
> >
> > **Re: (Weakness5) How is the performance when using fine-tuned MLLMs for reranking? Is it better than reranking in a zero-shot manner for tasks that involve single-modal queries?**
> >
> > We thank you for the insightful discussion. We believe fine-tuning MLLMs for reranking is a promising solution to push the effectiveness further in all the tasks. We hypothesize that there is task- or domain-specific knowledge in each task which is not leveraged by the zero-shot rerankers. That is, the relevance matching signals between queries and documents in News, Misc. and fashion domains are varied. The retrieval effectiveness on the tasks involving single-modal queries may be dominated by such domain-specific knowledge. A better prompt to assist MLLMs to conduct domain-specific relevance matching (e.g., clearly define what is relevance in News, Misc, and Fashion) or instruction fine-tuning is a future direction worth exploring. We have added this discussion (see lines 505--511 in our revised manuscript).
> >
> > **Re: (Weakness6) More qualitative cases are suggested to be provided to better illustrate the effects, such as how modality-aware hard negative mining alleviates modality bias.**
> >
> > We thank you for the insightful suggestions. We have appended the negative samples with incorrect modality (due to modality bias from MLLM retrievers) in Figure 2 (Appendix). Note that the negative samples with incorrect modality denote the retrieved candidates ranked higher than positive samples (as discussed in lines 199--200). Observing those negative samples with incorrect modality, they are semantically similar to the query but are not the user desired modality from the instruction. In addition, we conduct the full modality accuracy measurement on the full M-BEIR dataset and observe that the low modality accuracy (especially in NVEmb) mainly comes from the task of text--image retrieval, where the top-1 retrieved candidates are text rather than image. We have added the case study discussion of Figure 2 (see lines 214--215 in the revised manuscript) and the full measurement of  modality accuracy in our revised manuscript (see Appendix Table 11 in the revised manuscript) .

---

> > > ### Comment · Reviewer_ESAe · 2024-11-26
> > >
> > > I greatly appreciate the detailed responses and revised manuscript provided for each of the weaknesses from the original review.  I think, with the new, deeper investigation requested by me and other reviewers, the current revised paper has been significantly strengthend.  In light of this, I will positively update my review score to 6.

---

> ### Author Response · Authors · 2024-11-26
>
> We really enjoy the discussion and thank you for providing the valuable suggestions to improve our manuscript.

---

### Official Review · Reviewer_yjMm · 2024-10-31

**Soundness:** 2
**Presentation:** 2
**Contribution:** 3
**Rating:** 6
**Confidence:** 4

**Summary:**

This paper explores a more practical scenario in information retrieval, namely general multimodal retrieval. The author proposes a new fine-tuning pipeline and re-ranking method to improve the performance of specific retrieval tasks. However, the organization of the experimental results of this paper is rather coarse, and the performance presented makes me concerned about its generalization and robustness. It is recommended that the author further improve the presentation of results and methods. However, it is undeniable that the research topic can bring contribution and inspiration to the multimodal community.

**Strengths:**

1. The topic explored by the authors is a great one, namely general multimodal retrieval.
2. The method proposed by the author is effective to a certain extent.
3. The authors describe the experiment in detail and I think it is reproducible.

**Weaknesses:**

1. Unfortunately, the authors are not the first to use MLLM for re-ranking. It is recommended that the author faithfully investigate related research, e.g., [1]. The authors should narrow down the scope of the study, i.e., composed image retrieval.
2. The author's claim in line 184 should be verified experimentally using either very large k or very small k.
3. Line 306: The author's fine-tuned batch size for the two types of models is different. Will this affect the effect of mining negative samples?
4. I personally think that CLIP is a product from 3 years ago, and more advanced models should be used, such as BLIP or an improved version of CLIP. Moreover, the comparative experiment lacks experimental results of some recent methods that solve specific tasks, and it is impossible to quantitatively feel the advantages of the proposed MLLM-based method. The current organization is more like a complete ablation experiment.
5. From the results in Table 4, the full version re-ranking results do not seem to be optimal, which makes it difficult to explain the effectiveness of the method's continuous fine-tuning on some tasks. The improvement margin of its full version seems to be even worse than M^hard, which makes it hard to convince me.
6. From the studied results, the author expects to achieve universal retrieval, but as far as the experimental results (e.g., Table 7) are concerned, the proposed method will reduce the performance in some benchmarks. This makes me concerned about the generalization and robustness of its method.

[1] MERLIN: Multimodal Embedding Refinement via LLM-based Iterative Navigation for Text-Video Retrieval-Rerank Pipeline

**Questions:**

See weaknes.

---

> ### Author Response · Authors · 2024-11-20
>
> Many thanks for your comments and feedback. We discuss your raised points in the following.
>
> **Re: (Weakness1) Unfortunately, the authors are not the first to use MLLM for re-ranking. It is recommended that the author faithfully investigate related research, e.g., [1]. The authors should narrow down the scope of the study, i.e., composed image retrieval.**
>
> We thank you for pointing out the valuable reference, MERLIN[1]. We would consider this work as query rewriting or pseudo relevance feedback[2] in dense retrieval rather than prompt MLLMs for zero-shot reranking. First, the reference paper uses the embeddings from Google Multimodal API for both retrieval and reranking, which is a close-source model. Thus, we are not certain whether the API models used for reranking have been fine-tuned for retrieval tasks or not, while in our zero-shot reranking, we use LLaVa-Next, which is only fine-tuned on instructions without learning the tasks of retrieval. Nevertheless, we believe the approach in MERLIN can be further added to our retriever, UniEmb, to refine the query embedding with multiple search iteration to improve the final retrieval effectiveness, which is also a promising future direction and we will cite the reference paper in our future work. Also, we agree that we should fully investigate the related work for MLLM reranking to clarify our contribution (see our Related Work in the revised manuscript). Specifically, our approach is more close to prompt instruction fine-tuned models for zero-shot reranking[3][4], which has been previously explored in text ranking with LLMs but as far as we know, has not been comprehensively explored yet in multimodal retrieval with multimodal LLMs.
>
>
> [1] [MERLIN: Multimodal Embedding Refinement via LLM-based Iterative Navigation for Text-Video Retrieval-Rerank Pipeline.](https://aclanthology.org/2024.emnlp-industry.41/)
>
> [2] [Pseudo Relevance Feedback with Deep Language Models and Dense Retrievers: Successes and Pitfalls.](https://dl.acm.org/doi/10.1145/3570724)
>
> [3] [Is ChatGPT good at search? investigating large language models as re-ranking
> agents](https://aclanthology.org/2023.emnlp-main.923.pdf)
>
> [4] [A setwise approach for effective and highly efficient zero-shot ranking with large language models](https://arxiv.org/abs/2310.09497)
>
>
> **Re: (Weakness2) The author's claim in line 184 should be verified experimentally using either very large k or very small k.**
>
> We thank you for the helpful suggestions and agree that a clear explanation on the claim in line 184 should be given. In this paper, we directly borrow the previous hard negative mining study on text retrieval training from [1][2], which shows suboptimal retrieval effectiveness when mining hard negatives from too large and small top-$k$ candidates. Additionally, we also examine the retrieved results from our $M^{\text{rand}} (NVEmb)$ (which we use for hard negative mining for $M^{\text{hard}} (NVEmb)$) on the training set. Furthermore, we observe that the top-1 accuracy is above 50% in many query sets, such as WebQA, OVEN and INFOSEEK. This means when setting $k=1$, above 50% of the mined negatives are false negative while setting large $k$, the mined negatives would be similar to randomly sampled negatives from the corpus. We have added the explanation to our manuscript (see lines 203--205 in the revised manuscript). We agree that the optimal $k$ is case dependent and it is worth conducting a more comprehensive study on hyperparameter search in multimodal retrieval training, which we leave for future work.
>
> [1] [APPROXIMATE NEAREST NEIGHBOR NEGATIVE CONTRASTIVE LEARNING FOR DENSE TEXT RETRIEVAL](https://openreview.net/forum?id=zeFrfgyZln)
>
> [2] [NV-Retriever: Improving text embedding models with effective
> hard-negative mining.](https://arxiv.org/pdf/2407.15831)
>
> **Re: (Weakness3) Line 306: The author's fine-tuned batch size for the two types of models is different. Will this affect the effect of mining negative samples?**
>
> We thank you for the insightful question. We set different batch size at the two stages is because at the second stage, we pair each query with additional hard negative; thus, the total size of in-batch negatives is $2 \cdot |\mathcal{B}| - 1$ compared to the total size of in-batch negatives at first stage $|\mathcal{B}| - 1$, where $|\mathcal{B}|$ is batch size. Thus, to make the total size of  in-batch negatives the same at both stages, we use $2 \times$ larger batch size at first stage compared to the second stage. We revised our manuscript to clarify the reason behind the choice (see lines 211--215 in the revised manuscript).
>
> Continue...

---

> > ### Author Response · Authors · 2024-11-20
> >
> > **Re: (Weakness4) I personally think that CLIP is a product from 3 years ago, and more advanced ...**
> >
> > We thank you for the helpful suggestion. We agree that we should compare with other advanced models. Please see the general response, where we report the effectiveness of BLIP[1] fine-tuned on M-BEIR dataset, MagicLens[2] and E5-V[3].
> >
> > [1] [UniIR: Training and Benchmarking Universal Multimodal Information Retrievers](https://www.ecva.net/papers/eccv_2024/papers_ECCV/papers/11927.pdf)
> >
> > [2] [MagicLens: Self-Supervised Image Retrieval with Open-Ended Instructions](https://icml.cc/virtual/2024/poster/33731)
> >
> > [3] [E5-V: Universal Embeddings with Multimodal Large Language Models](https://arxiv.org/pdf/2407.12580)
> >
> > **Re: (Weakness5) From the results in Table 4, the full version re-ranking results do not seem to be optimal, which makes it difficult to explain the effectiveness of the method's continuous fine-tuning on some tasks. The improvement margin of its full version seems to be even worse than M^hard, which makes it hard to convince me.**
> >
> > We thank you for the comments. To clarify, the main purpose of continuous fine-tuning is to improve our multimodal retriever’s effectiveness on text-only retrieval, which is one of the most common retrieval scenarios. When comparing our final model UniEmb and $M^{\text{hard}}$(NVEmb) in Table 1, we also observe that UniEmb almost tie $M^{\text{hard}}$(NVEmb) in M-BEIR overall accuracy but cannot outperform in all the retrieval tasks while UniEmb significantly outperform $M^{\text{hard}}$(NVEmb) in text-only retrieval. Our ablation study in Table 6 also demonstrates the importance of continuous fine-tuning to reach superior retrieval effectiveness on both multimodal and text-only retrieval. We will revise our manuscript to make the contribution more clear.
> >
> > **Re: (Weakness6) From the studied results, the author expects to achieve universal retrieval, but as far as the experimental results (e.g., Table 7) are concerned, the proposed method will reduce the performance in some benchmarks. This makes me concerned about the generalization and robustness of its method.**
> >
> > We thank you for the comments. We want to clarify that with our proposed embedding model training approach, we have yielded state-of-the-art effectiveness on both multimodal and text-only retrieval benchmarks. Thus, UniEmb is a generalized and robust retrieval model across diverse tasks and modalities. As for the results on both Table3, 4 and 7, it is the exploration of prompting MLLMs to conduct zero-shot reranking. The important finding in these experiments is that MLLM can be a strong zero-shot raranker for the more challenging tasks where user queries are in multimodal format, especially such retrieval training data is far less and harder to collect than the large-scale text--image pairs used in CLIP and BLIP. We consider this finding a positive sign to further improve universal multimodal retrieval in such challenging scenarios.

---

> > > ### Comment · Reviewer_yjMm · 2024-11-25
> > >
> > > Thank you for your detailed responses. Your responses solved most of my concerns, but some details need to be refined especially the experimental organization. But it is not important. I will raise my score to 6. I hope the authors can further improve the quality of the manuscript.

---

> > > > ### Author Response · Authors · 2024-11-25
> > > >
> > > > We thank you for the valuable feedbacks, which help improve our manuscript. We'll continue improving the manuscript to make the experiment section more clear and well organized.

---

### Official Review · Reviewer_RfVM · 2024-11-04

**Soundness:** 3
**Presentation:** 2
**Contribution:** 3
**Rating:** 6
**Confidence:** 5

**Summary:**

In this paper, the authors focus on developing multimodal LLMs (MLLMs) for universal multimodal retrieval. Instead of directly fine-tuning MLLM-based bi-encoder retrievers with instructions as a guide on multiple multimodal retrieval tasks, the authors propose modality-aware hard negative mining and continual text-to-text retrieval fine-tuning to address the modality bias problem from MLLMs. Furthermore, the authors explore to prompting MLLMs as zero-shot rerankers, which can further boost retrieval accuracy in complex retrieval tasks. Extensive experimental results demonstrate the effectiveness of the proposed method.

**Strengths:**

1)	This paper explores the use of multimodal LLMs for universal multimodal retrieval, which is an interesting and potential research topic for the retrieval field.
2)	In this paper, the authors explore universal multimodal retrieval from three aspects, including modality-aware hard negative mining, continual text-to-text retrieval fine-tuning, and prompting MLLMs as zero-shot rerankers. These components are designed to improve the effectiveness of MLLM-based universal multimodal retrieval, while maintaining strong text-to-text retrieval capability.
3)	Extensive results demonstrate the effectiveness of the proposed method.

**Weaknesses:**

1)	The details of the proposed method are not clear and confusing. In sec. 4.1.1, it seems random negatives are denoted as c_i^{\times} symbol, while the generated triplets use c_i^{\times} again, does it mean positive sample or random negative?
2)	In sec. 4.2, the authors prompt LLaVa-Next for reranking, while not introducing the detailed MLLMs adopted in sec. 4.1. Are there two multimodal LLMs needed for the proposed method?
3)	It will be better to provide the framework figure for the whole method.
4)	In Table 4, it is not clear which results denote the reranking results, it seems the last row does not outperform the fifth row.
5)	As a retrieval method, the efficiency of retrieval time and memory is as important as effectiveness. The authors should analyze the efficiency of the unimodal retrieval mechanism.

**Questions:**

please try to address the weaknesses.

---

> ### Author Response · Authors · 2024-11-20
>
> Many thanks for your comments and feedback. We discuss your raised points in the following.
>
> **Re: (Weakness1) The details of the proposed method are not clear and confusing. In sec. 4.1.1, it seems random negatives are denoted as c_i^{\times} symbol, while the generated triplets use c_i^{\times} again, does it mean positive sample or random negative?**
>
> We thank you for pointing out the unclarity in our manuscript. To clarify, in the setting of training with random negatives, we pair each query in a mini batch with a positive; i.e., $((inst_1,q_1), c_1^+), ((inst_2,q_2), c_2^+), \cdots \((inst_{|\mathcal{B}|}, q_{|\mathcal{B}|}), c_{|\mathcal{B}|}^+\)$. While conducting contrastive training using Eq (1), we have in-batch documents from all the positive documents $\mathcal{D} = (c_1^+, c_2^+,  c_{|\mathcal{B}|}^+ )$. Thus, all the documents in $\mathcal{D}$ except for $c_i^+$ are considered randomly sampled negatives for $(inst_i,q_i)$. As for hard negative mining, we pair each query with a positive and hard negative $((inst_i,q_i), c_i^+, c_i^-)$. Thus, in the setting of hard negative mining, we have in-batch documents from all the positive and negative documents $\mathcal{D} = (c_1^+, c_1^-, \cdots, c_{|\mathcal{B}|}^+, c_{|\mathcal{B}|}^-)$. Similarly, all the documents $D$ except for $c_i^+$ are considered negatives for $(inst_i,q_i)$, including one hard negative $c^-_i$ and all the positives and negatives from other queries as random negatives.  We have revised our manuscript to make it more clear (see lines 195--215 in the revised manuscript).
>
> **Re: (Weakness2) In sec. 4.2, the authors prompt LLaVa-Next for reranking, while not introducing the detailed MLLMs adopted in sec. 4.1. Are there two multimodal LLMs needed for the proposed method?**
>
> We thank you for the suggestions. To clarify, we adopt the same MLLM backbone, LLaVa-Next. While retrieval, we explore fine-tuning MLLM as a retriever while reranking, we directly prompt the pre-trained MLLM as a zero-shot reranker. We have revised our manuscript (see lines 160--161 and lines 235--237) and Figure 1 in the revised manuscript to make it more clear.
>
> **Re: (Weakness3)  It will be better to provide the framework figure for the whole method.**
>
> We thank you for the suggestions. We have revised our Figure 1 to illustrate the whole framework of our paper, including MLLM retriever fine-tuning and prompt-and-reranking with pre-trained MLLMs (see Figure 1 in the revised manuscript).
>
>
> **Re: (Weakness4) In Table 4, it is not clear which results denote the reranking results, it seems the last row does not outperform the fifth row.**
>
> We thank you for the clarifying question. In Table 4, the first column denotes the retrieval result for each multimodal retriever (from MagicLens to our UniEmb retriever) and the second column denotes the zero-shot reranking result upon the top-10 candidates from each retriever. Thus, when comparing the first and second column, we can see that the zero-shot reranker improves over all the retrievers. We revised Table 4 to make it more clear to readers.
>
> **Re: (Weakness5) As a retrieval method, the efficiency of retrieval time and memory is as important as effectiveness. The authors should analyze the efficiency of the unimodal retrieval mechanism.**
>
> We thank you for the insightful suggestions. We measure the index storage required for the 5.6M document from the M-BEIR datatset. As for retrieval latency, we measure the latencies of query encoding and vector search. For query latency, we randomly sample 100 queries from each test query pool in the 16 M-BEIR tasks and measure per query encoding and vector search latency with a batch size of 1. Since query encoding latency is varied with query length, we report the latency at 1th, 50th and 99th percentiles. We have added the measurement and detail in Appendix A.2 in the revised manuscript.
>
> |Retriever |storage (GBs)  |Latency(ms) |Latency(ms)|
> | ----------- | ------------- | ------ | ------ |
> | |**Index** | **Encoding** (1st / 50th / 99th perc.)| **Vector search** |
> |  $\text{CLIP}_{\text{SF}}$| 16| 26 / 27 / 39|6|
> |  UniEmb  |86| 81 / 194 / 203| 33|

---

> > ### Comment · Reviewer_RfVM · 2024-11-26
> >
> > Thanks for the detailed response from the authors. These responses have addressed most of my questions, therefore, I will keep the original rating.

---

> > > ### Author Response · Authors · 2024-11-26
> > >
> > > We thank you for providing the constructive suggestions to improve our manuscript!

---

### Author Response · Authors · 2024-11-20

We thank all reviewers for providing the valuable insights and suggestions. We want to highlight the several changes in our revised manuscripts according to reviewers’ suggestions.
1. We revised Figure 1 to illustrate our approaches, including universal multimodal retriever fine-tuning and prompting-and-reranking with multimodal LLMs.
2. We added one more paragraph in Related Work to discuss prompting multimodal LLMs for reranking and compare the related work suggest by reviewers.
3. Universal multimodal retrieval is a relatively new but important task toward more general search. Although most prior work only focuses on limited retrieval tasks, we agree with reviews’ request to conduct a more comprehensive comparison with recent multimodal retrievers. In this revision, we add BLIP[1] fine-tuned on M-BEIR dataset into our main experiments and also we compare with the recent public multimodal retrievers, MagicLens[2] and E5-V[3] in some subtasks of M-BEIR.

**Table:M-BIER Local Eval**
||Dataset|UniEmb |BLIP[1]|MagicLens[2] |E5-V[3] |
| ----------- | ------------- | ------ | ------ |------ |------ |
|1. text --> image|MSCOC|82.7| 79.7| 68.5| 75.8|
|2. text --> text|WebQA|96.6|80.0| 47.9| 84.8|
|4. image --> text|MSCOC|91.0| 89.9| 17.4|83.4|
|5. image --> image|NIGHTS|32.7| 33.0|14.1|26.7|
|7. (image,text) --> image|FashionIQ|26.0|29.2|13.8|8.4|
|7. (image,text) --> image|CIRR|53.0|52.2|37.5|30.1|

Above Table reports the comparisons on the M-BEIR subtasks, which are used in E5-V paper (except for WebQA and Nights). Note that the evaluation setting is different from the setting in Table 1 in our paper since the compared models mainly focus on specific tasks rather than the tasks of universal multimodal retrieval. Here we use M-BEIR **local evaluation setting**, where retrieval is conducted only among the corpus consisting of less than 1M candidate documents with single modality from each subtask independently while in our paper Table 1, we use the more challenging **global evaluation setting**, where retrieval is conducted on the merged corpus consisting of 5.6M documents in diverse modality from all the 16 subtasks. Also note that while our reproduced E5-V results for MSCOCO is very close to the number reported in [1], we notice that there is a gap between the numbers in FashionIQ and CIRR since the corpus of the two tasks in M-BEIR is significantly larger than the ones used in [1]. For example, the corpus only contains 2K images in CIRR used in [1] while the corpus contains 21K images from M-BEIR CIRR task.

**Table: M-BIER Global Eval**
||UniEmb  |BLIP[1]|MagicLens[2] |E5-V[3] |
| ----------- | ------ | ------ |------ |------ |
|All|52.7| 45.5|5.8| 11.5|
|Single Modality Qry|56.1| 50.4| 8.1|14.6|
|Multi Modality Qry|47.0| 37.3|2.0| 6.3|

**Table: CIRCO Eval**
||Retrieval | Rerank|
| ----------- | ------ |------ |
BLIP[1]| 26.6| 36.1|
MagicLens[2]| 24.9| 32.4|
E5-V[3]|19.1| 31.0|
UniEmb|32.3| 39.9|

Above reports their overall retrieval effectiveness on the M-BEIR global evaluation dataset and we report their effectiveness (retrieval and after reranking with our zero-shot reranker) on CIRCO dataset. Note that, our variant LLaVa-P can be considered the fine-tuned version of E5-V[1] since in LLaVa-P, we borrow the idea from E5-V[1] to generate summarized embedding (see prompt in Table 12 Appendix) and further fine-tuned on M-BEIR training data.

[1] [UniIR: Training and Benchmarking Universal Multimodal Information Retrievers](https://www.ecva.net/papers/eccv_2024/papers_ECCV/papers/11927.pdf)

[2] [MagicLens: Self-Supervised Image Retrieval with Open-Ended Instructions](https://icml.cc/virtual/2024/poster/33731)

[3] [E5-V: Universal Embeddings with Multimodal Large Language Models](https://arxiv.org/pdf/2407.12580)

---

### Author Response · Authors · 2024-11-25

We thank all reviewers' helpful suggestions and discussion to help us improve our manuscript. Please let us know if there is any other concern or question.

---

### Meta-Review · Area_Chair_oSW4 · 2024-12-11

**Metareview:**

This paper explores trained and zero-shot versions of mixed-modality retrieval with help from MLLMs for reranking. The authors show effectiveness through extensive experiments. Some concerns exist about novelty (in using MLLM for re-ranking), experiments and writing/clarity (method details, relationship to prior work), but these are addressed in the rebuttal phase. All reviewers recommend acceptance, although mostly lean only very weakly positive.

**Additional Comments On Reviewer Discussion:**

Concerns were addressed during the rebuttal, as noted by the reviewers

---

### Decision · Program_Chairs · 2025-01-22

Accept (Poster)